# De novo evolved interference competition promotes the spread of biofilm defectors

Marivic Martin[1,*], Anna Dragoš[1,*], Theresa Hölscher[1,*], Gergely Maróti[2], Balázs Bálint[3], Martin Westermann[4] & Ákos T. Kovács[1]

Biofilms are social entities where bacteria live in tightly packed agglomerations, surrounded by self-secreted exopolymers. Since production of exopolymers is costly and potentially exploitable by non-producers, mechanisms that prevent invasion of non-producing mutants are hypothesized. Here we study long-term dynamics and evolution in *Bacillus subtilis* biofilm populations consisting of wild-type (WT) matrix producers and mutant non-producers. We show that non-producers initially fail to incorporate into biofilms formed by the WT cells, resulting in 100-fold lower final frequency compared to the WT. However, this is modulated in a long-term scenario, as non-producers evolve the ability to better incorporate into biofilms, thereby slightly decreasing the productivity of the whole population. Detailed molecular analysis reveals that the unexpected shift in the initially stable biofilm is coupled with newly evolved phage-mediated interference competition. Our work therefore demonstrates how collective behaviour can be disrupted as a result of rapid adaptation through mobile genetic elements.

---

[1] Terrestrial Biofilms Group, Institute of Microbiology, Friedrich Schiller University Jena, Jena 07743, Germany. [2] Institute of Biochemistry, Biological Research Centre, Hungarian Academy of Sciences, Szeged 6726, Hungary. [3] Seqomics Biotechnology Ltd., Mórahalom 6782, Hungary. [4] Electron Microscopy Center, Jena University Hospital, Jena 07743, Germany. * These authors contributed equally to this work. Correspondence and requests for materials should be addressed to Á.T.K. (email: atkovacs@dtu.dk).

Biofilms, consisting of densely packed single- or multi-species communities embedded in self-produced slimy polymers, represent the most common microbial life form[1-3]. Several recent studies have shown that the spatial structure of biofilms has a major impact on competition and cooperation among microbes and drives evolutionary changes within microbial communities (reviewed in refs 4,5). One particularly well-studied example used static cultures of *Pseudomonas fluorescens*, where an oxygen gradient led to the emergence of a new wrinkly (W) phenotype that secretes polysaccharides and forms a biofilm at the air–liquid interface[6,7]. Interestingly, biofilms formed by W undergo a premature collapse caused by the incorporation of another phenotype into the biofilm without sharing the metabolic costs of exopolymer production[8]. This scenario of biofilm collapse reflects the phenomenon known as 'tragedy of the commons', which happens due to invasion by non-cooperators and depletion of an overly-exploited resource (in this case the exopolymer)[9].

How often the 'tragedy of the commons' happens in other biofilm communities remains an open question in sociomicrobiology. Several studies suggest that exopolymer production cannot easily be exploited by non-producing defectors[10,11]. Such robustness of cooperation-based biofilm formation is often explained by limited sharing of matrix components[10–12], the low costs of polymer production[11], the spatial assortment of cells in biofilms[13] or even the intrinsic nature of certain matrix components that are exclusively shared between mother and daughter cells[14]. Although the key principles of certain non-producer exclusion mechanisms are becoming clear, competition experiments involving producers and non-producers are usually conducted over short timescales[11–14], leaving a window of opportunity for unexpected evolutionary scenarios[15]. Data from various bacterial models suggest that defectors can leave a fingerprint on the evolution of social strains and promote the evolution of novel cheating-suppression mechanisms[16]. These can be linked to lowering the cost of cooperation by the wild-type (WT) cells[17]. Selection can also work to the advantage of the non-producers, which can evolve better exploitation skills[15,17]. In extreme cases, cooperators can be *de novo* selected from the population of cheats[18]. In general, long-term scenarios in socially heterogeneous populations of microbes are still very difficult to predict.

In this manuscript, we study the long-term social dynamics of co-cultures comprising matrix producer and non-producer strains using the widespread soil bacterium *Bacillus subtilis*. *B. subtilis* forms thick, robust structures at the air-liquid interface (pellicle) facilitated by two crucial secreted compounds: an exopolysaccharide, Eps (encoded by *epsA-O*), and a protein component, TasA (encoded by *tapA-sipW-tasA*). In a standing culture, driven by oxygen limitation, matrix-producing strains form pellicles[19]. Strains lacking either one or both matrix components cannot form robust biofilms at the air–liquid interface and they barely colonize the liquid surface[20]. Moreover, strains producing only one of the components are able to complement each other and form a WT-like pellicle[20]. This strongly suggests that both matrix components secreted by producers are freely shared with non-producers and could therefore be exploited by non-producing mutants.

Here we show that on a short timescale, *B. subtilis* matrix non-producers have a tremendous disadvantage in co-culture with the WT. We further demonstrate how unexpected adaptive events involving mobile genetic elements can shift the social dynamics in the population and reduce biofilm formation.

## Results

**Biofilm non-producers are outcompeted from mixed pellicles.** A positive result in a complementation assay of *B. subtilis* Δ*eps* and Δ*tasA* biofilm mutants suggests that both key biofilm components, Eps and TasA, can be shared (Fig. 1a)[20]. We therefore predicted that the double mutant Δ*eps*–Δ*tasA*, which cannot form a pellicle in monoculture[20], would still be able to incorporate into the pellicle when co-cultured with the WT. To test our hypothesis, we mixed WT and Δ*eps*–Δ*tasA* strains in a 1:1 ratio and allowed the pellicle to form (see Methods). The final ratio of the WT to the Δ*eps*–Δ*tasA* strain was assessed by two alternative methods: antibiotic marker based colony forming unit (c.f.u.) counts (Fig. 1a) and fluorescence microscopy (here, GFP and mKATE2 producing WT and Δ*eps*–Δ*tasA* mutants were used, respectively, or we used the same strains with swapped fluorescent markers; Fig. 1b,c). Surprisingly both c.f.u. assay and microscopy indicated a dramatic advantage of the WT over

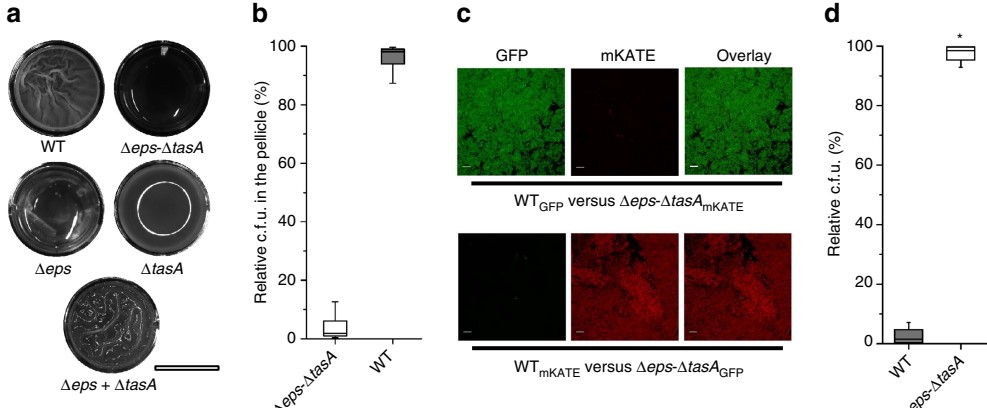

**Figure 1 | Matrix non-producer strain does not incorporate into the wild-type pellicle.** (**a**) Pellicle biofilms of wild-type (WT) *Bacillus subtilis* 168 and its mutant derivatives were recorded using an Axio Zoom microscope equipped with a black and white camera. Scale bar, 1 cm. The reflection of the light source can be observed in the Δ*tasA* culture. (**b**) Pellicle competition assay between Δ*eps*–Δ*tasA* and WT (*n* = 25). (**c**) Confocal microscopy images of pellicle biofilm from culture initially consisting of a 1:1 ratio of WT and Δ*eps*–Δ*tasA* (upper) and cells with swapped fluorescence protein labels (lower). Scale bars, 10 μm. (**d**) Planktonic culture competition assay between Δ*eps*–Δ*tasA* and WT (*n* = 25). Boxes represent Q1–Q3, lines represent the median, and bars span from max to min. The experiments were independently repeated at least three times. * indicates that the relative c.f.u. is significantly higher than the relative c.f.u. of Δ*eps*–Δ*tasA* ancestor in the pellicle (in panel **b**).

Δeps–ΔtasA; the latter was almost completely outcompeted from the pellicle formed by the WT (Fig. 1a–c).

The incorporation success of Δeps–ΔtasA into the pellicle was positively dependent on its initial frequency (Pearson's correlation coefficient $r = 0.74$; Supplementary Fig. 1A). Specifically, the mutant showed increased pellicle incorporation, up to $23 \pm 1.9\%$, which also correlated with decline in total c.f.u. of the pellicle, but only when its initial proportion was $> 50\%$, while at initial frequencies $< 50\%$, the pellicle incorporation ranged from 0.11 to 5% (Supplementary Fig. 1A).

To understand if the availability of nutrients influenced the ability of non-producers to incorporate into the pellicle, the competition assay was repeated using a medium in which the broth and other components were doubled ($4 \times SG$, see Methods). It was observed that the Δeps–ΔtasA strain could incorporate better in richer medium (Supplementary Fig. 1B). Using $2 \times SG$, the incorporation ability of the Δeps–ΔtasA strain was 2% (mean; $n = 5$; s.d. $= 1.37$), while in $4 \times SG$ medium pellicle incorporation increased to 6.72% (mean; $n = 5$; s.d. $= 3.40$). Importantly, the starting ratios in these competition assays were identical (46.72% Δeps–ΔtasA; mean; $n = 5$; s.d. $= 14.4$), therefore, the possibility of the initial frequency influencing these results could be excluded.

Finally, to ensure that the above result was caused by a mechanism that is specific to biofilm conditions and not simply caused by a growth defect of the Δeps–ΔtasA strain, WT versus Δeps–ΔtasA competition experiments were also performed in planktonic cultures where oxygen distribution is more homogenous and no fitness benefits from biofilm formation are to be expected[11,13,21]. In planktonic culture, Δeps–ΔtasA had a strong fitness advantage over the WT (Fig. 1d) that is likely due to the release of the mutant from the metabolic costs of Eps and TasA production[13], as also indicated by its higher growth rate in planktonic culture conditions (Supplementary Fig. 1C). We therefore concluded that a specific mechanism prevents incorporation of the Δeps–ΔtasA mutant into B. subtilis pellicles.

**The ratio of non-producers increases during co-evolution.** For the investigation of long-term dynamics in B. subtilis WT and Δeps–ΔtasA mutant co-culture over time, a serial transfer experiment was conducted in conditions promoting pellicle formation (see Methods). During the evolution experiment, two transfer methods were applied: in transfer method A, the disrupted biofilm suspension was used directly for the inoculation of fresh medium; in transfer method B the disrupted biofilm suspension was heat-treated, thereby selecting only spores for the inoculation (for detailed description see Methods). Method B was chosen to select for individuals that successfully went through the entire biofilm life cycle.

The ratio of WT to the Δeps–ΔtasA mutant was monitored by selective plating of frozen stocks prepared at different timepoints from the experiment, from the 2nd up to the final (10th) transfer. Despite the initial incorporation failure of Δeps–ΔtasA into the pellicle (Fig. 1a–c), its representation in certain populations of transfer method B was observed to increase dramatically over longer timescales. Remarkably, in parallel populations where transfer method B was applied (that is, selection for spores), the fraction of the Δeps–ΔtasA mutant was considerably higher after the 10th transfer than at the start of the experimental evolution in all but two replicates (Fig. 2b). Importantly, with several exceptions (in replicates 2 that remained relatively stable WT:Δeps–ΔtasA ratio over time and replicate 4 that showed an outlying outburst of Δeps–ΔtasA at passage 8), the percentage of the Δeps–ΔtasA mutant was increased in each successive passage of these populations (Fig. 2b). The values rose to $> 30\%$ in

general and to a maximum of around 80% after the 10th transfer of replicate 5. Also, in one out of five parallel populations that were transferred by method A, the fraction of Δeps–ΔtasA was slightly higher after the 10th transfer compared to that at the beginning of the evolution experiment (Fig. 2a).

**Non-producers evolve to better incorporate into the pellicle.** To further investigate the evolutionary phenomena involved in improved performance of Δeps–ΔtasA in the evolved biofilm population, single clones of both the WT and Δeps–ΔtasA mutant were isolated from three randomly chosen populations after the 10th transfer where an increase of Δeps–ΔtasA in the pellicle was observed (replicates 3, 4 and 5 from transfer method B) (Fig. 2b). All evolved populations and single clones that were further analysed (or genetically modified) in this study are listed in Supplementary Table 2. For clarity, we refer to evolved matrix producers (WT strains) as eMP and to the evolved matrix non-producers (Δeps–ΔtasA) as eNMP.

First, to understand which of the co-cultured strains evolved to facilitate better incorporation of the mutant into the pellicle, a series of pellicle competition assays were performed. Competition assays revealed that all but one tested eNMP strains from populations B410m and B510m, and one isolate from population B310m, could increase their fraction within the pellicles as compared to their ancestor when co-cultured with the ancestor WT (Fig. 3a,b). This result was confirmed by both c.f.u. assay (Fig. 3a) and fluorescence microscopy (Fig. 3b). Moreover, the ancestor Δeps–ΔtasA performed even worse when co-cultured with the eMP strains compared to its performance against the WT ancestor (Fig. 3). Therefore, the eMPs completely suppressed the ancestral Δeps–ΔtasA.

The performance of three selected eMP and eNMP representatives (one from each evolved population) against the WT ancestor was additionally determined by calculating the selection rate coefficient. All eMPs showed a positive selection rate and their relative c.f.u. in the pellicle was significantly higher that 50% (Supplementary Fig. 3). However, the ancestor Δeps–ΔtasA and eNMPs had negative selection rates, which indicates poor performance during competition with the ancestor WT. Nevertheless, the ancestor Δeps–ΔtasA strain showed the poorest performance (selection rate value of $-3.36$) and all the eNMPs, B310mA, B410mB and B510mC, revealed improved performance compared to the ancestor mutant strain, with selection rate values of $-2.59$, $-1.14$ and $-2.25$, respectively.

Finally, the eNMPs were challenged with the eMPs selected from the corresponding populations (that is, B310mA versus B310wtA or B410mB versus B410wtB). We noticed that the eNMPs from population B310m exhibited a slight decrease in pellicle incorporation compared to their pellicle incorporation when in competition with the ancestor WT (Fig. 3). Overall the eMPs performed better at suppressing the eNMPs as compared to the WT ancestor, however, certain eNMPs from populations B410m and B510m still displayed significantly improved incorporation whether competing against the evolved or ancestor WT (Fig. 3). On the basis of these competition assays, we conclude that evolutionary changes in the Δeps–ΔtasA mutant, rather than the WT, resulted in the improved performance of the non-producers in mixed pellicles.

It was further revealed that the incorporation success of the eNMPs did not depend on their initial frequency. Competition assays with different starting ratios of the WT ancestor to each of the eNMPs (B310mA, B410mB and B510mC) revealed that the eNMPs exhibited higher levels of pellicle incorporation regardless of their starting frequency (Supplementary Fig. 1D–F). B310mA, B410mB and B510mC showed an average pellicle incorporation percentage of

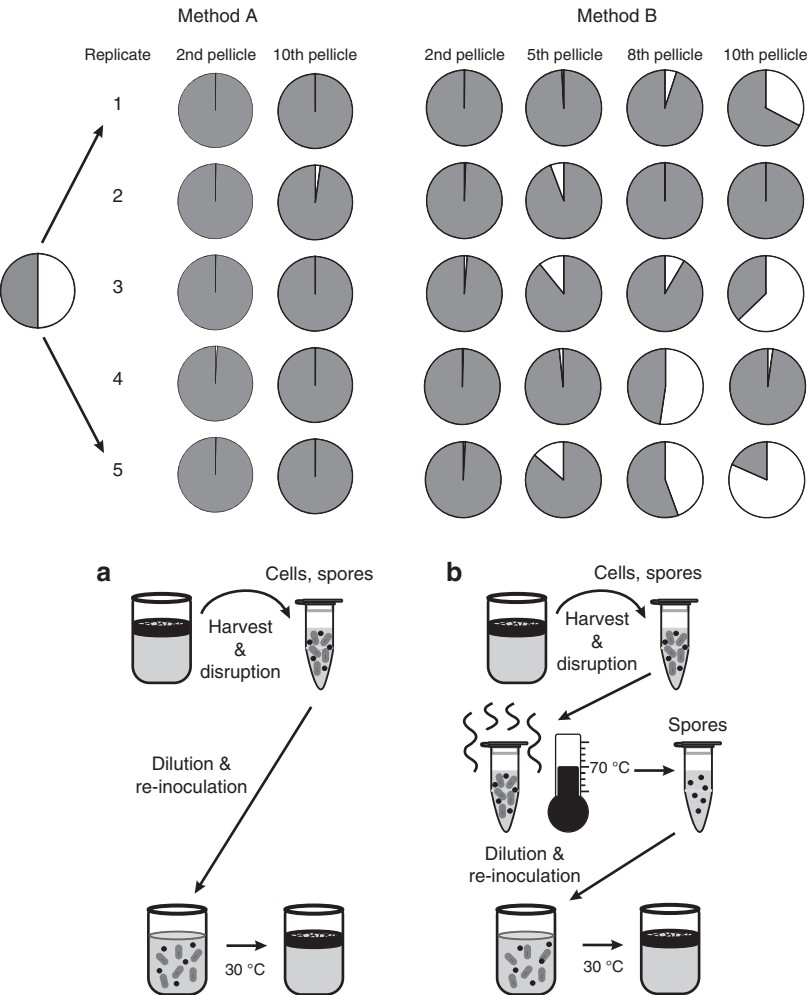

**Figure 2 | Fraction of WT and Δeps–ΔtasA strains changes during pellicle serial transfer.** In transfer method (**a**), the pellicle was harvested, disrupted (see Methods) and the suspension of spores and cells was directly used for inoculation of new medium. After harvest and disruption, the cells in transfer method (**b**) were heat-treated at 70 °C for 15 min to eliminate all vegetative cells. The resulting spore suspension was used to inoculate new medium and was incubated for 2–3 days to allow for pellicle formation. For transfer method (**b**), the gradual change of wild-type (grey) and Δeps–ΔtasA (white) ratio during the evolution experiment is also presented. The ratio was followed during evolution experiments using the disrupted pellicle suspension directly. In addition, the ratio was redetermined from frozen glycerol stocks for the second, eighth and tenth pellicles.

9.13% (Pearson's correlation coefficient $r = 0.17$), 18% ($r = 0.22$) and 27% ($r = 0.16$), respectively (Supplementary Fig. 1D–F).

Further experiments also revealed that, in contrast to the ancestor mutant (Supplementary Fig. 1B), the incorporation percentage of the eNMP B310mA was not affected by doubling the concentration of resources in the medium (Supplementary Fig. 1B); the incorporation of the evolved B310mA was 9.54% ± 3.04% in 2 × SG medium and 9.88% ± 2.04% in 4 × SG. These results suggest that the incorporation efficiency of the eNMPs might be driven by a different mechanism from that in the ancestor Δeps–ΔtasA.

**Incorporation of eNMPs decreases biofilm productivity.** A productivity assay was performed to understand the effect of increased incorporation of the eNMPs on the biofilm productivity and to compare the productivity of the eMPs relative to the ancestor WT. Productivity was measured by weighing the whole biomass of the pellicle and is represented as relative productivity compared with the ancestor WT (that is, ancestor WT productivity = 1).

As expected, the productivity of the mixed pellicle consisting of the WT ancestor and mutant ancestor was very similar to the productivity of the WT ancestor grown alone, indicating that the presence of the ancestor Δeps–ΔtasA did not affect the biofilm productivity (Fig. 4). This result agrees with our results showing that ancestor Δeps–ΔtasA was almost completely outcompeted from the pellicle (Fig. 1a–c). In contrast, the productivity of pellicles containing both the ancestor WT and the eNMPs was lower than the productivity of the monoculture WT (productivity values <1), indicating that the population was negatively affected overall when eNMPs were present (Fig. 4). Interestingly, the eMPs in monocultures (B310wtA, B410wtB and B510wtC) had higher productivity than the WT ancestor (Fig. 4; Supplementary Fig. 4A). Finally, we examined the productivities of the evolved pairs with common evolutionary histories (B310wtA + B310mA; B410wtB + B410mB; and B510wtC + B510mC). For all three pairs, the productivity of the mixed pellicles was lower than the productivity of the ancestor WT; however, these differences were statistically significant only for the pairs B410wtB + B410mB and B510wtC + B510mC (Fig. 4). Nevertheless, in all combinations, the eNMP + eMP productivities were significantly lower than the corresponding eMP productivity, indicating reproducible

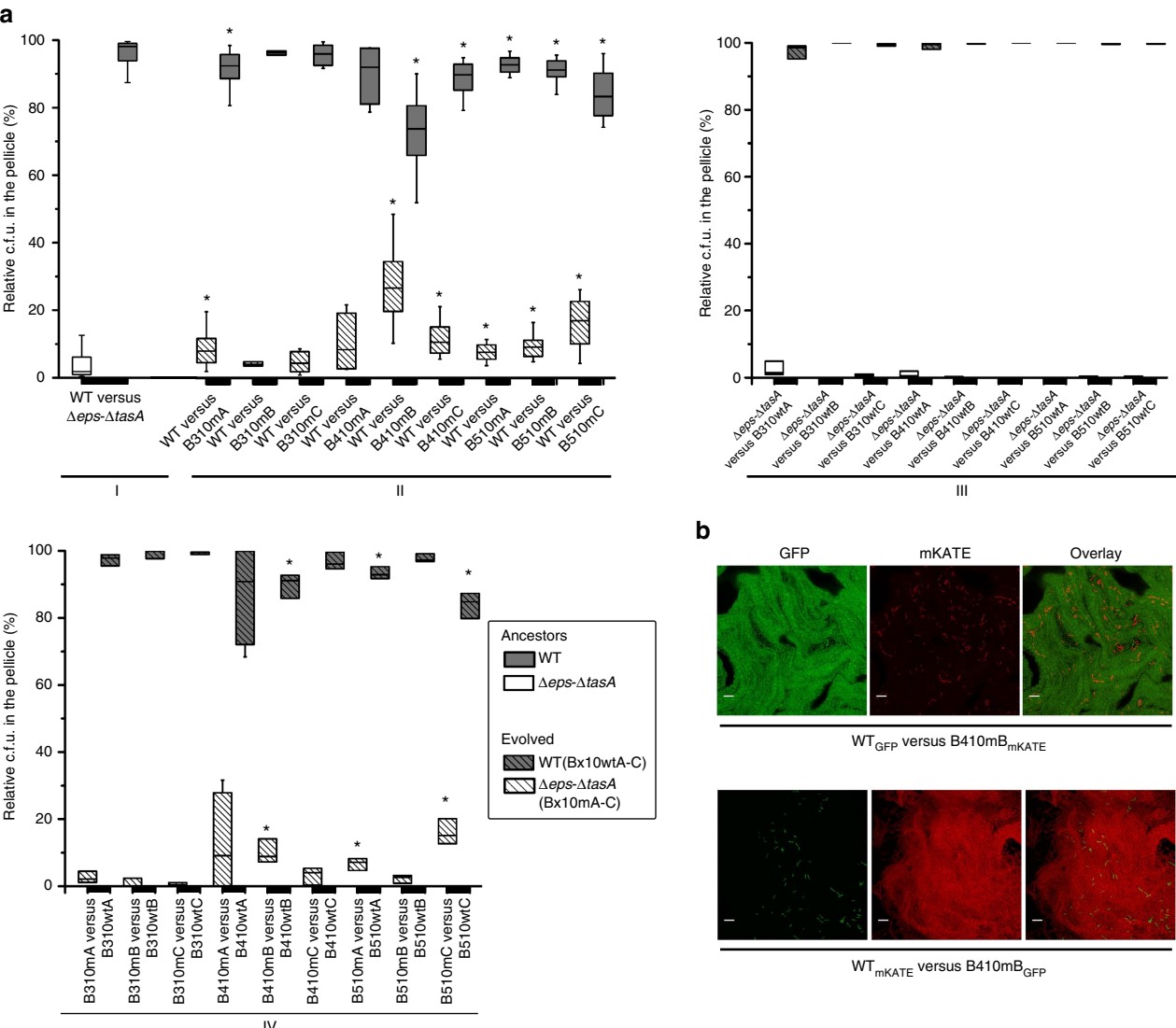

**Figure 3 | Pellicle competition assay.** (**a**) WT ancestor and Δeps–ΔtasA mutant ancestor (I); and WT ancestor and eNMP strains 310mA, 310mB, 310mC, 410 mA, 410mB, 410mC and 510mA, 510mB, 510mC (II). On a separate panel (above right): pellicle competition assay between WT ancestor and Δeps–ΔtasA mutant ancestor, eMP strains (three isolates per population, as in **a**) and Δeps–ΔtasA mutant ancestor (III). On a panel below: pellicle competition assay between WT ancestor and Δeps–ΔtasA mutant ancestor, eMP strains and eNMP strains sharing evolutionary history (IV). Boxes represent Q1–Q3, lines represent the median, and bars span from max to min. Each competition assay was replicated in parallel with the ancestor WT versus Δeps–ΔtasA combination at least twice. * in sections II and IV indicate that the relative c.f.u. are significantly different from the relative c.f.u. of WT versus Δeps–ΔtasA ancestor competition. (**b**) Confocal microscopy images of pellicle biofilms (left) including swapped fluorescence marker proteins (right). Scale bars, 10 μm.

negative effects of the eNMPs on the productivity of the entire evolved population (Fig. 4). In addition, the pellicles formed by the populations from sequential evolutionary timepoints showed overall decreases in productivity in evolutionary time (Supplementary Fig. 4B), presumably caused by the increasing frequency of eNMPs in pellicles (Fig. 2b).

These results show that although matrix producers evolved a higher productivity, higher incorporation of the coevolved matrix non-producers into the pellicle eventually decreased the overall population productivity.

**eMPs and eNMPs contain multiple SNPs in prophage elements.** To understand the genetic basis of the observed evolutionary dynamics, the genomes of three eMP and three eNMP populations separated from the 10th transfer of method B cultures (replicates 3, 4 and 5), where the frequency of non-producers was observed to increase during evolution (either gradually or periodically), were subjected to high-

throughput sequencing (Supplementary Table 2). The genomes of corresponding three single isolates of eMPs (B310wtA, B410wtB and B510wtC) and three eNMPs (B310mA, B410mB and B510mC) from those populations were also sequenced. In addition, the genomes of the WT ancestor and the Δeps–ΔtasA ancestor were resequenced to screen for any single SNPs that emerged before the evolution experiment during standard stock preparation and laboratory procedures. The sequencing of six populations (eMPs B310wt, B410wt and B510wt and eNMPs B310m, B410m and B510m) and six single isolates (B310wtA, B410wtB, B510wtC, B310mA, B410mB and B510mC) revealed multiple single-nucleotide polymorphisms (SNPs) exclusively accumulated in three distinct sites on the chromosome compared to the ancestors: two prophage-like regions previously described as prophage-like element 5 and prophage-like element 6 (ref. 22), and the SPβ prophage region (Supplementary Data 1; Fig. 5a,b). In population B310 there were 617 SNPs, while in populations B410 and B510 the number of SNPs exceeded 1000. More than

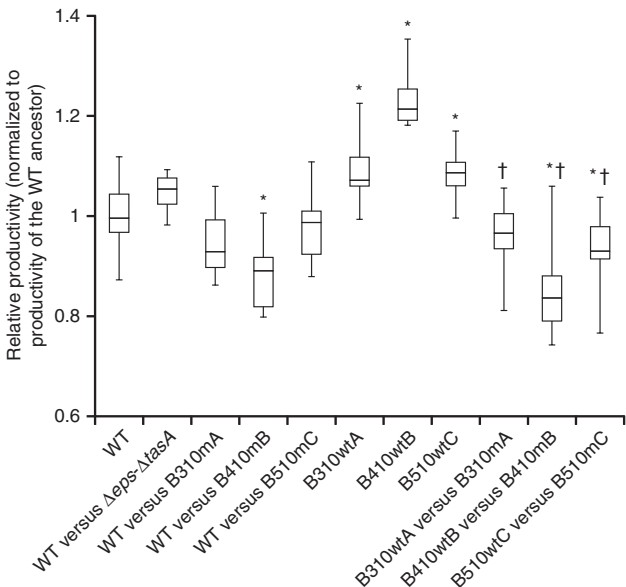

**Figure 4 | Productivity assay based on biofilm biomass.** Productivity of Δeps–ΔtasA ancestor and eNMPs B310mA, B410mB and B510mC, respectively, in co-cultures with WT ancestor, monocultures of eMPs B310wtA, B410wtB and B510wtC, and co-cultures of eNMPs with corresponding eMPs compared to the WT ancestor ($n = 10$; t-Student; two-tail $P < 0.05$). Boxes represent Q1–Q3, lines represent the median, and bars span from max to min. Each WT versus co-culture/eMP comparison was replicated at least twice. *—productivities significantly different from the WT ancestor. †—productivities significantly lower from the corresponding eMP cultures.

50% of SNPs detected in single isolates overlapped with SNPs found in the corresponding populations (Supplementary Data 1; Fig. 6a,b).

Further analysis of the sequenced genomes revealed that there was also a large parallel mutational overlap between the evolutionarily unrelated populations and single isolates (regardless of WT or Δeps–ΔtasA background; Supplementary Data 1; Fig. 6a,b). For visual representation of this overlap we produced a windowed-average identity score for the alignment of the entire 134 kbp SPβ region of the evolved strains/populations to the ancestral SPβ (Supplementary Fig. 5A). A global analysis of SNPs from all six single isolates showed that the majority of SNPs represented synonymous substitutions (58%), 17% were non-synonymous but evolutionarily conserved (that is, similar; Blosum62 matrix score ≥ 1), 9% were evolutionarily non-conserved and non-synonymous (Blosum62 matrix score ≤ 0), and the remaining 16% of the substitutions were located in non-coding regions (Supplementary Fig. 5B). We also compared the distributions of SNPs in the eMPs and eNMPs by analysing the functions of affected genes. We observed that eMPs accumulated more SNPs than the corresponding eNMPs, especially in genes related to the toxin production and secretion (Supplementary Data 1; Supplementary Fig. 5D,E). However, most of the affected genes belonged to the unknown function category.

More detailed analyses of the sequencing data on the evolved strains suggested duplications of certain genome fragments and genome rearrangements compared to the ancestors. Duplications were indicated by the increased sequencing coverage within the SNP-containing regions (Supplementary Fig. 6A) and the striking pattern of SNP frequencies (Supplementary Data 1; Supplementary Fig. 6B), which was confirmed by PCR and Sanger sequencing of the particularly highly-mutated SPβ fragment (2,178,034–2,179,407) from the genomic DNA of B310mA and

B310wtA (Supplementary Fig. 6C). Interestingly, the PCR product obtained from B310mA gave a clear chromatogram with all SNPs present, whereas B310wtA showed a heterogeneous chromatogram with double peaks in the positions of SNPs, one peak coming from an ancestor-like base and the other from the evolved-like base (Supplementary Fig. 6C). In addition, the SPβ fragments could still be amplified by PCR even after deletion of the original SPβ region from the chromosomes of B410mB and B510mC (Supplementary Fig. 6D–F). The identification of genome rearrangements was made after de novo assembly of sequencing reads into contigs (Fig. 6c). All of the predicted rearrangements involved sequences belonging to prophage-like elements 5 and 6 and various SPβ fragments, and included the exact regions where multiple SNPs accumulated (Fig. 6c). The presence of two randomly selected rearrangements (contig type 1 and type 4) was confirmed by PCR to occur exclusively in the evolved strains; it did not occur in the ancestor WT or ancestor Δeps–ΔtasA (Supplementary Fig. 6G). Altogether, we conclude that the emergence of multiple SNPs in all evolved strains (both eMPs and eNMPs) was linked to duplications and rearrangements within prophage elements in the B. subtilis genome. It is important to note that the mutation frequencies of the ancestor and the evolved strains were similar, as confirmed using fluctuation assays (Supplementary Fig. 5C). The obtained mutation frequencies were comparable to previously reported data for other B. subtilis strains[23], suggesting that the ancestor strains used here were not hypermutators. Moreover, when the same ancestor strain was evolved for ~350 generations in emulsion droplets, 60 SNPs and short deletions were identified (Eisha Mhatre and Á.T. Kovács, unpublished data).

**Hybrid SPβ prophage shows lytic activity towards the ancestors.** Rearrangements involving SPβ prophage regions have previously been described as a result of the hybridization of SPβ with another B. subtilis phage, phi3T (ref. 24). A hybrid form of SPβ can undergo spontaneous excision from the chromosome to form a pseudolysogen, or it can enter a lytic cycle leading to active phage-particle release[24]. To verify whether the eMPs and eNMPs in the present study spontaneously released phage particles into the medium, phages were precipitated from the supernatants of cultures of selected evolved strains and of the WT ancestor (as a negative control) and visualized by transmission electron microscopy.

No phage particles could be detected in the precipitate obtained from the WT ancestor, which was in line with previous findings[25]. When the WT ancestor was grown in the presence of the prophage-inducing agent mitomycin C, PBSX-like phage particles were detected in its supernatant, which again reproduced previous results[26] (Supplementary Fig. 7A). However, even in the absence of mitomycin C, the evolved strains B410mB and B410wtB released two types of phage particles—PBSX-like particles with a small head and a rigid tail (assignment based on ref. 26), and SPβ-like particles with a big head and a longer, flexible tail (assignment based on an image provided by Vladimir Lazarevic, Hôpitaux Universitaires de Genève, Switzerland, personal communication; Fig. 7a; Supplementary Fig. 7A). The addition of mitomycin C to B410mB and B410wtB cultures resulted in a dramatic increase in the number of SPβ-like phage particles in the culture supernatants (Supplementary Fig. 7A). SPβ-like particles could not be detected in the supernatant of B310mA$^{SPβ −}$ cultures, but were still present in B410mB$^{SPβ −}$ cultures, which corresponded well with the results of molecular analysis, which indicated successful deletion of SPβ from strain B310mA$^{SPβ −}$ but not from B410mB$^{SPβ −}$ (Supplementary Figs 6D–F and 7A).

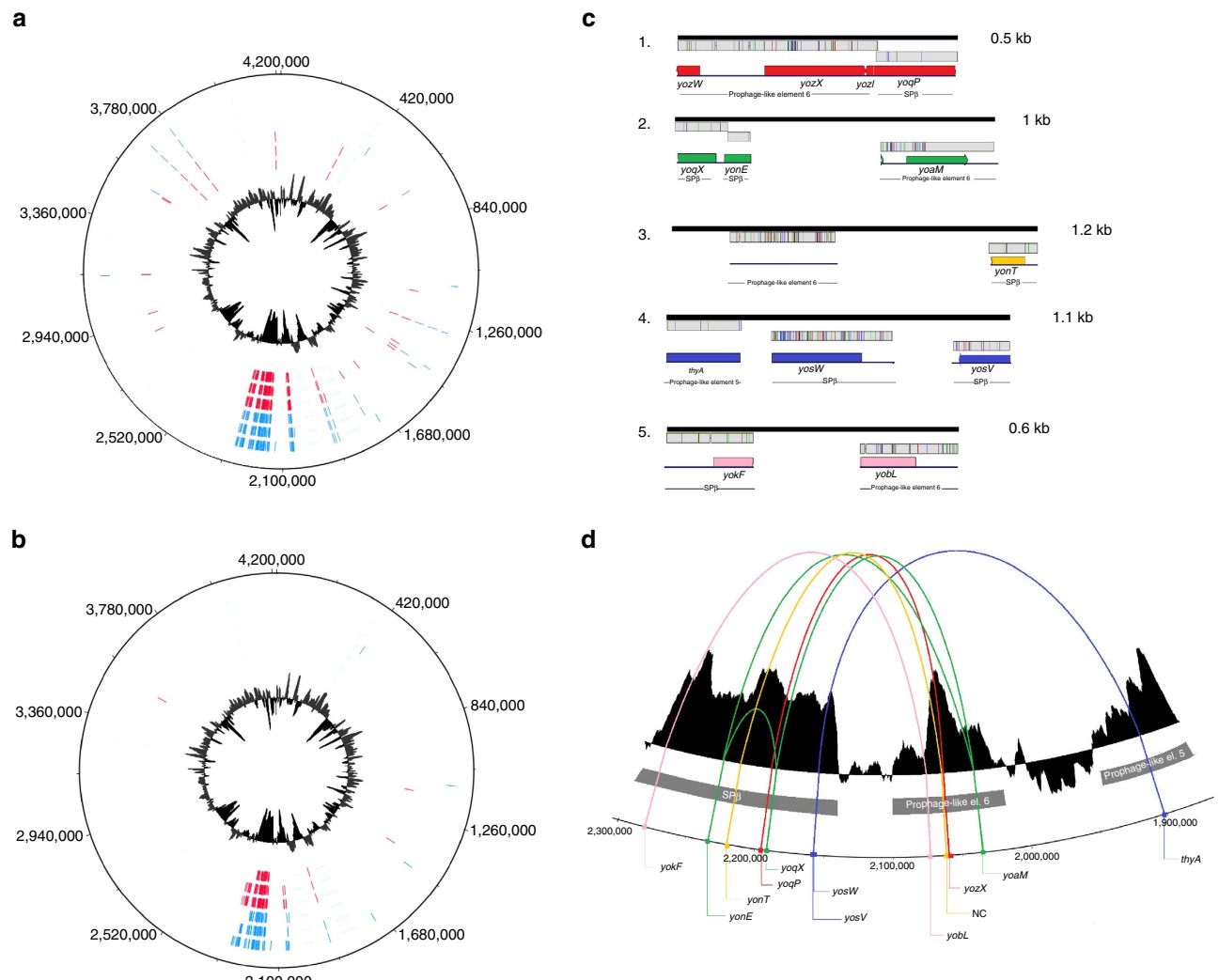

**Figure 5 | Multiple SNPs and genome rearrangements are detected in the evolved strains.** Genome wide distribution of SNP loci across the evolved *B. subtilis* populations (**a**) and single isolates (**b**) generated using the DNAPlotter tool[56]. (**a**) SNPs from six populations are presented on separate tracks; starting from the outside and moving towards the inside: B310wt, B410wt, B510wt (all shown in blue), B310m, B410m, and B510m (all shown in red). (**b**) SNPs in six isolates are presented on separate tracks; starting from the outside and moving towards the inside: B310wtA, B410wtB, B510wtC (all shown in blue), B310mA, B410mB, and B510mC (all shown in red). Internal black circles represent GC profiles. (**c**) Genome rearrangements predicted for the evolved strains based on bioinformatics analysis of the sequencing data. Black lines represent *de novo* assembled contigs with their sizes indicated on the right, grey boxes show aligned fragments of the reference genome (GenBank accession number AL009126), and stripes show SNP positions. Visualization was performed using Geneious software[57]. Corresponding genes (or non-coding regions) in the reference *B. subtilis* genome are shown below the alignment with different coloration used for each type of rearrangement. (**d**) DNA plotter graphical representation of predicted genome rearrangements (zoom view on the SPβ and prophage-like element 6 and 5 regions). The black histogram represents the GC profile, where three clusters of low-GC content directly correspond to SPβ, prophage-like element 6 and prophage-like element 5. Distantly located genes that participated in rearrangements and became neighbours in the evolved strains are connected by arched lines (different colour used for each rearrangement) and are additionally marked with dots of the same colour. The coloration used for each rearrangement type corresponds to that in **c**.

Next, the lytic activity of the SPβ particles released by the evolved strains was tested against the ancestor strains. A series of plaque assays were performed where each strain served both as a supernatant donor and as potential prey. Neither of the ancestor strains (WT or Δeps–ΔtasA) showed lytic activity when serving as the supernatant donor, but they were both susceptible to the lytic activity of almost all supernatants of the evolved strains (Fig. 6b, Supplementary Fig. 7B). Strain B310mA^SPβ − performed exactly the same as the ancestors, showing no lytic activity but displaying susceptibility to all supernatants, including that of B310mA (from which it was derived; Fig. 6b; Supplementary Fig. 7B). Despite the fact that all evolved strains showed lytic activity and immunity, they could be differentiated into strong (for example, B410mB) and moderate levels (for example,

B510wtC) (Fig. 6b; Supplementary Fig. 7B). The lytic activity of the supernatants of all evolved populations was assessed, including all five populations from transfer method A and all five populations from transfer method B (Fig. 2). Strong lytic activity towards the ancestor WT strain was found exclusively in populations that showed an increased incorporation of non-producers into the pellicle following the evolution experiment, specifically population 2 from transfer method A, and populations 1, 3, 4 and 5 from transfer method B (Supplementary Fig. 7C). Further, populations that did not show increased incorporation of non-producers and lacked lytic activity towards the ancestor strains did not contain multiple SNPs within the SPβ regions, as confirmed by Sanger sequencing of the 2,178,034–2,179,407 genomic fragment.

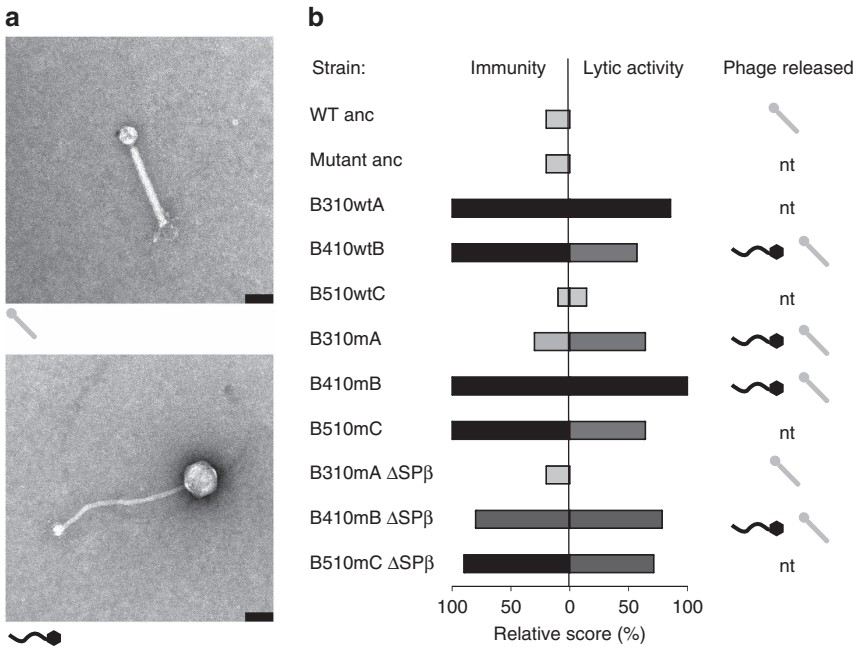

**Figure 6 | Lytic phage activity appears in the evolved strains.** (**a**) Electron micrographs of phage particles purified from *B. subtilis* supernatants. Upper image shows PBSX-like phage particle released by the ancestor WT only in the presence of mitomycin C, and by all evolved strains in the absence of mitomycin C. The lower image shows SPβ-like phage particle spontaneously released by all evolved strains tested, but not by the ancestor WT or by B310mA$^{SPβ-}$. Scale bars, 70 nm. (**b**) Results of plaque assays performed with the ancestor WT, ancestor Δeps–ΔtasA and all the evolved strains, where each strain served both as a supernatant donor and as a potential host. Each strain was given 1 immunity point when resistant to a given supernatant (excluding its own supernatant), and a number of lytic activity points when showing lytic activity towards a given host (excluding itself), depending on the strength of lytic activity. Specifically, the number of points was equal to a maximal log dilution factor where lytic activity was still present (for example, a lytic activity that can still be detected in a conditioned medium diluted to $10^{-3}$ but not $10^{-4}$ is denoted with 3 points). Obtained immunity and lytic activity scores were then divided by the maximum value of each and are presented as relative percentages. Darkness of the bars is proportional to immunity/lytic activity scores values. High immunity/lytic activity scores correlated with presence of SPβ-like particles (black in the right-hand column) isolated from the culture medium of those strains. 'nt' indicates media were not tested by transmission electron microscopy. The experiment was independently replicated four times.

Spontaneous phage release by the evolved strains and their lytic activity towards the ancestors suggested that higher incorporation of the eNMPs into pellicles may be the result of newly evolved interference competition. We therefore examined whether the ancestor mutant could acquire the evolved-like phenotype with higher incorporation pellicle properties through a single-phage transduction step. For the infection assay, a 1:1 mixture of the WT and Δeps–ΔtasA ancestors was introduced into standard 2 × SG medium supplemented with phage precipitate obtained from B410mB where the presence of SPβ-like phage particles was detected. After a single growth cycle, three colonies of the WT and three colonies of Δeps–ΔtasA were isolated and their acquired lytic activity towards the ancestor strains was assessed. Finally, pellicle competition assays were performed using the WT ancestor and the three Δeps–ΔtasA strains isolated from the infected population (ImA, ImB and ImC). As expected, the phage-infected strains behaved similarly to the evolved mutants, showing >4-fold (in the case of ImA) and >2.5-fold (ImB and ImC) increased incorporation rates into the pellicle compared with the ancestor mutant (Fig. 7).

**Phage release facilitates higher pellicle inclusion of eNMPs.** Finally, we asked whether the presence of an identical active phage variant in both producers and non-producers is sufficient to explain the higher incorporation of the eNMPs into pellicles. This was first tested by assaying the infected mutants (ImA, ImB and ImC) with the infected WT strains (Supplementary Fig. 8). No increased pellicle incorporation of the mutants was observed, indicating that higher incorporation of the mutants cannot be explained by a general increase of phage activity in the entire

population (Supplementary Fig. 8), but is due instead to subtle differences within phage elements of evolved non-producers and producers.

This was further confirmed by a fitness assay that involved Δeps–ΔtasA and WT strains with an isogenic evolved background. Isogenic evolved WT and mutant strains were obtained simply by introducing the Δeps–ΔtasA deletions into eMPs. Genome resequencing confirmed that the obtained Δeps–ΔtasA strains still contained the genetic background of corresponding eMPs (Supplementary Data 1). When the eMPs B310wtA, B410wtB and B510wtC were competed against their direct derivatives B310wtA$^{Δeps–ΔtasA}$, B410wtB$^{Δeps–ΔtasA}$, and B510wtC$^{Δeps–ΔtasA}$, respectively, a very low pellicle incorporation percentage of the mutants was observed, which was comparable to the performance of the ancestor Δeps–ΔtasA against the ancestor WT (Supplementary Fig. 8). As expected, competition assays with the WT ancestor revealed that the transformants had comparable incorporation probabilities to the eNMPs (Fig. 7). These results indicated that although producers and non-producers showed very similar general adaptation patterns involving major changes in mobile genetic elements, some of these changes were specific to the evolved non-producers, resulting in their improved incorporation into pellicles, most likely through an advantage in interference competition.

## Discussion

Stability of cooperative interactions can determine the performance of microbes in most medically and biotechnologically relevant situations[27–32]. In recent years, understanding of microbial group behaviours and the mechanisms that prevent

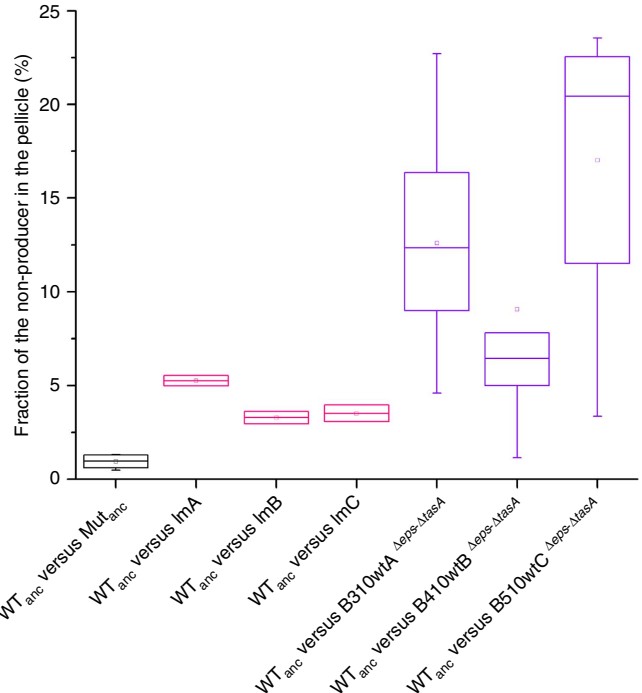

**Figure 7 | Evolved mutants and ancestor mutants hosting the evolved phage have increased incorporations into the pellicle.** Competition assay between the WT ancestor and $\Delta eps$–$\Delta tasA$ ancestor (control) ($n = 7$); between the WT ancestor and three single colonies obtained after transduction of the $\Delta eps$–$\Delta tasA$ ancestor with phage particles released by eNMP B410mB ($n = 2$); and between the WT ancestor and eMPs (B310wtA, B410wtB, and B510wtC) with deleted *eps* and *tasA* ($n = 10$; $n = 10$; $n = 7$, respectively). Boxes represent Q1–Q3, lines represent the median, and bars span from max to min. Each competition assay in parallel with the ancestor WT versus $\Delta eps$–$\Delta tasA$ was replicated at least twice.

spreading of non-cooperative mutants has become one of the key aims of sociomicrobiology. Long timescale evolutionary experiments have already demonstrated the evolutionary plasticity of social interactions in various bacterial models[18,33,34]. Here we describe a scenario where a biofilm matrix non-producer that is initially eliminated from the population increases its performance over longer timescales and evolves the ability to better incorporate into the biofilm. The evolution of improved invasion of biofilms by non-producers was previously observed by Zhang *et al.*[15]. They excluded the possibility of general adaptation being responsible for the changed social dynamics in biofilms since the evolved producer did not increase its performance towards the ancestor producer. In the present work, an increased selection coefficient and improved productivity of the evolved WT could be observed in monocultures; the same, unfortunately, could not be tested for the evolved mutants because of their inability to form pellicles in monocultures. We therefore hypothesize that the evolved increased-biofilm-incorporation-ability of the mutant was a side effect of extremely fast general adaptation of both producer and non-producers driven by mobile genetic elements. Interestingly, the improved incorporation of the non-producers into biofilms was not reproduced when both WT and non-producer strains had identical evolved genetic backgrounds (that of the evolved WT strains). This means that although the general adaptation pattern in the entire population was very similar, the non-producers are evolutionarily ahead of the producers and carry certain specific changes that allow their improved performance in

incorporation into biofilms formed by the evolved WT strains. We believe those specific differences are hidden within prophage elements of the evolved strains, and they could be revealed by *de novo* sequencing in the future.

In the ancestral population, the matrix non-producers ($\Delta eps$–$\Delta tasA$, which do not secrete two key matrix components Eps and TasA) can hardly incorporate into pellicle biofilms formed by the WT. This result was rather unexpected for two reasons: first, previous work demonstrated that both Eps and TasA are shared with non-producing strains[20], and, second, the production of at least one of those compounds (Eps) was proven to be costly and exploitable[13]. Although we did not study the competition mechanism in detail, a positive correlation between fitness, initial $\Delta eps$–$\Delta tasA$ frequency and resource availability suggests that in the ancestral population the growth of $\Delta eps$–$\Delta tasA$ is not only limited by the lack of oxygen, but also by carbon resources. We speculate that this is caused by a delay in surface co-colonization of $\Delta eps$–$\Delta tasA$, because the producer can partially privatize the matrix components. Since the WT is released from oxygen-limitation first, it can quickly deplete the remaining carbon resources, preventing further growth of the mutant. This model, however, awaits further studies.

The pellicle incorporation mechanism of evolved $\Delta eps$–$\Delta tasA$ does not depend on resource concentration or on the initial frequency of the mutant in the co-culture. It is likely that new antagonistic interactions involving infection and lysis of the ancestor WT by the evolved mutant delay surface colonization by the WT, giving the mutant a prolonged window of opportunity for co-colonization. A similar mechanism could play a role in the competition between the evolved mutant and the evolved WT, since the evolved WT strains spontaneously release phages into the medium and show a delay in pellicle formation.

How did the new lytic properties evolve? We believe that multiple rearrangements in the genomes of the evolved strains, combined with series of SNPs in regions that were rearranged, resulted in new lytic properties of the normally inactive domesticated SPβ prophage. Since this scenario was more likely to occur on sporulation treatment (that is, treatment method B), we suspect that the multiple heat-treatments involved in this treatment might have promoted phage activation[35] or even rearrangements of phage elements in the genome[36]. The accumulation of multiple SNPs and rearrangements resembles the previously reported evolutionary response of the *Streptococcus thermophilus* phage to the host's CRISPR system[37], however, no CRISPR/Cas has yet been identified in *B. subtilis*. Alternatively, rapid diversification within prophage regions combined with lytic induction may be a universal adaptive pattern of bacteria to a biofilm lifestyle, as it was previously also observed during experimental evolution of *Pseudomonas aeruginosa* biofilms[38]. Our work also demonstrates how such newly evolved phage warfare shifts social dynamics in the bacterial population in favour of biofilm non-producers. The dynamics of host-phage interactions is long studied in various experimental systems[39]. It was previously observed that lytic phages can shift the balance in competitive interactions by reducing the frequency of a winning partner[40], or impair biofilm formation ability as a trade-off for phage immunity[41]. We hypothesize that in the case of the *B. subtilis* pellicles, the disadvantage of matrix producers could originate from the degeneration of toxin/secretion-related genes in the evolved wild-types that in turn became less efficient competitors than the evolved mutants.

The improved fitness of the evolved WT strain in monoculture could be a direct result of the evolved spontaneous phage release. Normally, the excision of the SPβ prophage from the *B. subtilis* chromosome takes place before sporulation and allows recon-stitution of the *spsM* gene involved in spore polysaccharide

biosynthesis[42]. Sanchez-Vizuete et al.[43] demonstrated that removal of SPβ from the chromosome permanently restores spsM, resulting in increased biofilm thickness. We presume that frequent spontaneous excision of SPβ, or even pseudolysogeny (as demonstrated in ref. 24) in the evolved WT strains, could positively contribute to the biofilm productivity through spsM reconstitution. Excision of prophage from the host chromosome was recently linked to improved biofilm formation by Shewanella oneidensis facing cold stress[44]. Such a phage excision benefited the host through gene inactivation rather than reconstitution (as observed in ref. 43). Similar genetic switches triggered by prophage excision were also described in several other species (reviewed in ref. 45).

The B. subtilis SPβ prophage carries a bacteriocin-immunity system[46], several putative toxin–antitoxin systems[47] and cell wall hydrolases[48]. Several SPβ segments of >250 nucleotides exhibit >90% identity with B. subtilis chromosomal regions[25] promoting recombination events, especially in naturally competent strains. Not surprisingly, recent reports strongly indicate a key role of phage elements in rapid evolution of kin recognition mechanisms and antagonistic interactions between closely related, sympatric B. subtilis strains[49,50]. Accumulation of SNPs in the SPβ region was also observed in the evolution experiments of Overkamp et al.[51], where B. subtilis was kept in zero-growth conditions for 42 days. Among hundreds of SNPs discovered by Overkamp et al.[51], 80% overlapped with the SNPs reported in this study. In addition, most of the SNPs detected were synonymous and evolutionarily conserved, suggesting selection against loss of function. Recent reports show that even non-synonymous mutations can positively contribute to fitness[52,53]. This again suggests that mutations or rearrangements within phage elements can be a very important evolutionary force in B. subtilis, with a major impact on social interactions. Recently, the profound impact of prophages on the evolution of a pathogenic bacterium was experimentally demonstrated in P. aeruginosa biofilms[54], where the presence of phages resulted in strong selection against phage recognition elements (type IV pilus), at the same time enhancing parallel evolution[54]. Similar selective pressure could emerge after fast evolution of active SPβ variants in B. subtilis biofilms, resulting in striking parallelism in evolved populations of both WT and Δeps–ΔtasA bacteria.

Our work demonstrates how social dynamics in an initially very robust biofilm can be shifted by unexpected evolutionary events. We show that an adaptive genotype that is quickly tailored by mobile genetic elements can easily spread through horizontal gene transfer. The same adaptive path, although beneficial for the producer, became maladaptive in a mixed population where producers coexisted with non-producers.

## Methods

**Strains and cultivation conditions.** Supplementary Table 1 describes strains used in this study and construction of their mutant derivatives. Strain B. subtilis 168 hymKATE P$_{tapA}$-yfp was obtained by transforming the laboratory strain, B. subtilis 168, with genomic DNA from DL821 and selecting for MLS resistance. Subsequently, the created strain was transformed with genomic DNA from 168 hymKate and selecting on chloramphenicol resistance and for the loss of amylase activity. The Δeps and ΔtasA strains were obtained by transforming the 168 strains with genomic DNA isolated from DL1032 and specifically selecting for tetracycline or kanamycin resistance, respectively. The double mutant Δeps–ΔtasA was obtained by transforming 168 Δeps with genomic DNA from DL1032 and selecting on the kanamycin marker. The Δeps–ΔtasA hyGFP and Δeps–ΔtasA hymKate strains were obtained by transforming the Δeps–ΔtasA strain with genomic DNA obtained from 168 hyGFP and 168 hymKate, respectively. Deletion of epsA-O and tasA genes were confirmed with PCR using oligos described in Supplementary Table 3. Strains were maintained in LB medium (Lennox broth; Carl Roth, Germany), while 2 × SG medium was used for biofilm induction[55].

**Experimental evolution and competition assays.** Experimental evolution was performed using co-cultures of fluorescently labelled but otherwise WT and

Δeps–ΔtasA strains grown in 2 ml 2 × SG medium statically in a 24-well plate at 30 °C for 2–3 days. For transfer method A, the mature pellicles were harvested, mildly disrupted, and reinoculated after 100 × dilution. For transfer method B, the pellicles were additionally heat-treated after disruption and diluted × 20 during reinoculation. The sporulation frequency in the conditions applied in the evolution experiment was about 20% (Supplementary Fig. 2). To maintain similar selection bottlenecks in the two transfer methods, a fivefold lower dilution factor was used in transfer method B.

After the 2nd, 8th and 10th pellicle transfers, frozen stocks were preserved. Evolved populations or single isolates were isolated by selecting with appropriate antibiotics. Competition experiments were performed by mixing certain ratios of 100-fold diluted LB-pregrown cultures which were then incubated in static pellicle forming conditions for 3 days or in agitated planktonic cultures for 16 h. The numbers of c.f.u. of the inocula and the final cultures were determined on LB-agar plates containing selective antibiotics, incubated overnight at 37 °C. Prior the c.f.u. assays, pellicles were sonicated according to a protocol optimized in our laboratory (2 cycles each containing 12 × 1 s pulses at 20% amplitude with 1 s pause between the pulses), that ensured efficient disruption of biofilm clumps (as verified by microscopy) and therefore accurate total cell counts in the pellicles.

**Microscopy.** Bright field images of whole pellicles were obtained with an Axio Zoom V16 stereomicroscope (Carl Zeiss, Jena, Germany) equipped with a Zeiss CL 9000 LED light source and an AxioCam MRm monochrome camera (Carl Zeiss). The pellicles were also analysed using a confocal laser scanning microscope (LSM 780 equipped with an argon laser, Carl Zeiss) and Plan-Apochromat/1.4 Oil DIC M27 × 63 objective. Fluorescent reporter excitation was performed with the argon laser at 488 nm and the emitted fluorescence was recorded at 484–536 nm and 567–654 nm for GFP and mKate, respectively. To generate pellicle images, Z-stack series with 1 μm steps were acquired. Zen 2012 Software (Carl Zeiss) was used for both stereomicroscopy and CLSM image visualization.

**Productivity assay.** For productivity assays, pellicles were inoculated into 4 ml of 2 × SG medium placed in 35 mm-diameter Petri dishes and incubated for 3 days at 30 °C. Next, the medium fraction was removed, and pellicles were dried at 55 °C for 3 h. The dry biomass was determined on an analytical balance.

**Fluctuation assay.** To determine the mutation rate, single colonies were picked from LB-agar medium and cultivated for 18 h in LB broth at 37 °C. After 100-times dilution in 2 × SG medium, cultures (n = 10 for each strain) were subsequently cultivated for 18 h with vigorous shaking, and dilution series were plated on LB-agar medium to assay the frequency of streptomycin (50 μg ml$^{-1}$) resistant c.f.u. after 18–24 h at 37 °C.

**Genome resequencing and genome analysis.** Genomic DNA of selected populations or isolated strains was isolated using the EURex Bacterial and Yeast Genomic DNA Kit from cultures grown for 18 h. For the evolved population, single-end fragment reads were sequenced using a Life Technologies SOLiD 5500xl sequencer. Base-calling was carried out with the software provided by the supplier. All other downstream analysis steps were done in CLC Genomics Workbench Tool 7.0.4. Reads were length-filtered, keeping only ≥50 nucleotide long fragments. Mapping used only those reads that displayed ≥80% similarity to the reference genome (GenBank accession number AL009126) over ≥60% of the read length (meaning an alignment of ≥30 nucleotides having ≥24 identical matches). Non-specific reads were randomly placed to one of their possible genomic locations. Quality-based SNP and small in/del variant calling was carried out requiring ≥10 × read coverage with ≥20% variant frequency. Only variants suggested by good quality bases (Q≥20) were taken into account. Furthermore, mutations had to be supported by evidence from both DNA strands.

For single isolate strains, paired-end fragment reads (2 × 250 nucleotides) were generated using an Illumina MiSeq sequencer. Primary data analysis (base-calling) was carried out with MiSeq Reporter software (Illumina). All further analysis steps were done in CLC Genomics Workbench Tool 8.0.2. Reads were quality-trimmed using an error probability of 0.05 (Q13) as the threshold. Reads that displayed ≥80% similarity to the reference genome (GenBank accession number AL009126) over ≥80% of their read lengths were used in mapping. Non-specific reads were randomly placed to one of their possible genomic locations. Quality-based SNP and small In/Del variant calling was carried out requiring ≥40 × read coverage with ≥20% variant frequency. Only variants supported by good quality bases (Q≥20) were taken into account and only if they were supported by evidence from both DNA strands. Selected genomic regions were validated by Sanger sequencing (GATC Biotech, Konstanz, Germany) using oligos listed in Supplementary Table 3.

**Transmission electron microscopy analysis.** Selected bacterial strains were grown overnight in LB medium at 37 °C with shaking at 200 r.p.m. In the case of mitomycin-C-treated cultures, mitomycin C was added in late exponential phase to a final concentration of 0.5 μg ml$^{-1}$. Culture supernatants were collected, mixed at a 1:4 ratio with PEG-8000 solution (PEG-8000 20%, 2 M NaCl), incubated on ice for at least 90 min and finally centrifuged (20 min, 7,600 r.p.m.) to obtain

precipitate. The pellet was resuspended in 10% of the original supernatant volume in TBS solution (50 mM Tris-HCl, 150 mM NaCl, pH 7), incubated on ice for 90 min and centrifuged (20 min, 7600 r.p.m.). Supernatant was carefully transferred to clean Eppendorf tubes. Purified samples (100 µl) were adsorbed onto duplicate 400 mesh carbon-coated Cu grids (Quantifoil, Großlöbichau, Germany) for 2 min. Before use, the carbon grids were hydrophilized by 30 s of electric glow discharging. The grids were washed twice in distilled water and stained for 30 s with 1% uranyl acetate. Virus morphologies were examined using a Zeiss CEM 902A transmission electron microscope (Carl Zeiss AG, Oberkochen, Germany). At least 20 images were taken per sample at different magnifications using a 1k FastScan CCD-Camera (camera and software from TVIPS, Munich, Germany).

**Statistical analyses.** Statistical differences between two experimental groups were identified using two-tailed Student's $t$-tests assuming equal variance. Variances in the two main types of datasets (c.f.u. counts in competition assays and weight of biomass) were similar across different samples. One data point with a value greater than the mean plus 3 times the s.d. was removed from the dataset of $n > 10$ as an outlier. Normal distributions within the two main data types (biomass and c.f.u.) were confirmed by Kolmogorov–Smirnov ($P > 0.05$). No statistical methods were used to predetermine sample size and the experiments were not randomized.

**Data availability.** The genome resequencing data are available in Supplementary Data 1. The authors declare that all other relevant data supporting the findings of the study are available within the article and its Supplementary Information files, or from the corresponding author upon request.

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

## Acknowledgements

We are grateful to Christian Kost and Michael Brockhurst for helpful discussions. *B. subtilis* strain SPmini was kindly provided by Prof Tsutomu Sato (Research Center of Micro-Nano Technology, Hosei University). This work was funded by grant KO4741/2-1 from the Deutsche Forschungsgemeinschaft (DFG) within the priority program SPP1617 and a Startup Fund from Jena School for Microbial Communications (JSMC). MG was supported by GINOP-2.3.2-15-2016-00011 (EU Structural Funds). T.H. and A.D. were supported by International Max Planck Research School and Alexander von Humboldt foundation fellowships, respectively.

## Author contributions

A.T.K. conceived the project; M.M., A.D., T.H., and A.T.K. designed the research; M.M., A.D. and T.H. performed the research; M.M., A.D. and T.H. analysed the data; M.W. performed the electron microscopy; G.M. and B.B. performed and analysed genome resequencing; and M.M., A.D., T.H., and A.T.K. wrote the paper.

## Additional information

**Competing interests:** The authors declare no competing financial interests.

