## [Peer Review File · Nature Communications]

Reviewers' comments:

Reviewer #1 (Remarks to the Author):

This is an interesting study in which biofilm-matrix non-producers of *B. subtilis* evolved to more effectively integrate into pellicle biofilms generated by matrix producers. This improved ability to colonize pellicles was associated with reduced population productivity in mixes of one of three evolved non-producers with the ancestral producer. Additionally, increased pellicle colonization, involved a form of phage-related interference competition unique to evolved non-producers in that it was not exhibited by engineered non-matrix-producing mutants of evolved matrix producers. These central results are interesting, novel (to our knowledge) and seemingly sound. However, there are several substantial points that need to be addressed.

Major points

Pellicle disruption

Is it known that the method of biofilm dispersion used prior to dilution plating completely disperses all biofilm cells? Might biofilm cells remain substantially clumped after the dispersion step, but leading to underestimates of population size? Also, might the degree of cell-cell cohesion itself have evolved in the experiment for either producers (MPs for matrix producers), non-producers (NMPs) or both, thus potentially affecting the relative accuracy of dilution plating for the two categories over evolutionary time? Is this potentially a problem for interpreting fitness estimates? If not, please explain why.

Pellicle productivity

In lines 212-215, the authors make a general statement that the evolved NMPs (eNMPs) reduce productivity of pellicles. However, this is only in the context of mixes with the WT ancestor, relative to that ancestor alone. Moreover, from Fig. 4 it seems that only one of the eNMPs has a significant negative effect individually, and no test for a general effect across all three eNMPs seems to be reported. But more importantly, the most relevant treatment would seem to be eNMPs mixed with evolved MPs (eMPS), rather than eNMPs mixed with unevolved MPs because the latter two categories are evolutionarily disconnected from one another. The critical question seems to be whether eNMP + eMP mixes have lower productivity than NMP + MP mixes. If not, then eNMPs do not lower productivity in the context of the populations in which they co-evolved.

Related to this, do the authors have productivity estimates for mixes of eNMPs and eMPs at intermediate time points shown in Fig. S2? If so, it would be nice to see if pellicle productivity correlates negatively with eNMP-frequency in pellicles across all replicates over time.

Because productivity was measured as biomass production, can the authors say anything about whether how much of the increased productivity of the eMPs is due to increased cell #s vs. to increased matrix production?

Mutation rates

The authors do not specify the relevant unit for the mutation rate estimates in Fig. S4C but should do so. Is it per genome, per gene (*rpsL*)? If the latter, the mutation rate would be extremely high, higher in fact than most hyper-mutator strains of *E. coli*. In wildtype *E. coli* strains this rate has been estimated as 5×10^{-10} per target (Levin et al. 2000 GENETICS, 154, 985-997), which would indicate that this *B. subtilis* strain had a mutation rate six orders of magnitude higher than the rate of *E. coli* (which is around 10^{-10} per bp per generation; Lee et al. 2012 PNAS) if the same unit is being considered.

Minor points.

Nomenclature. The authors frequently use 'community/ies' when the correct term is 'population/s' because all variation in this study is intraspecific.

Fig. S1. It would be helpful to show lines corresponding to slope 1.0 in panels A, C, D and E to facilitate rapid interpretation of the data-point distributions.

Reviewer #2 (Remarks to the Author):

The article suggest that non-pellicle forming *B. subtilis* are excluded from the pellicle, but subsequently evolved to overcome this exclusion by the lytic action of prophage being used as biological weapons. This is a potentially very exciting result, making an important contribution to our understanding of microbial social evolution in biofilms. There is an impressive body of work here, but I had trouble with some of the interpretation of the data.

1. Most significantly, the data as it stands does not demonstrate that the mutant evolves to be more efficient at exploiting the wildtype biofilm. I appreciate that the mutant increases in frequency over 10 transfers, but it may have done this in the absence of genetic change. The key experiments are the competition experiments using the ancestral and evolved mutants and wildtype (Fig 3). These do not show that mutant has become more efficient at exploiting the evolved wildtype (Fig 3A IV), or even that the evolved mutants are better at exploiting the wildtype ancestor (Fig 3A II) but this is probably inevitable given that only 3 evolved mutants and wildtype were tested. Specifically, at least 1 out of 3 mutants doesn't show increased exploitation ability in both cases.

2. I was also very confused with some of the logic. The data convincingly shows that lytic phage are released in evolved population (both wildtype and ancestor). However, the authors argue that this phage release helps exploitation of the biofilm by the mutant in the (co)evolved populations. However, the data in Fig S7 which uses isogenic evolved WT-mutant pairs, both of which release the phage, show no increase in mutant fitness relative ancestral pairs which do not release the phage

3. Sequencing. The number of SNPs seems extremely high to me (<1000 in some cases in less than 100 generations), even given the prophages. Just curious.

4. Please be explicit that there are no consistent differences in SNP distribution between evolved mutants and wildtypes

5. Taken together, the results suggest that the evolution of *B. subtilis* in these conditions results in parallel genomic and phenotypic evolution the mutant and wildtype, with phage release being a notable fitness enhancing trait against ancestral strains. While interesting (and previously observed in other systems, as noted by the authors), this does not seem to have anything specifically to do with the evolution of social exploitation.

6. On a semantic note, I don't think there is any evidence that the wildtype excludes mutants – it may simply be that the mutants can't exploit the wildtype pellicle. I think this is quite an important distinction both conceptually and mechanistically.

Reviewer #3 (Remarks to the Author):

The manuscript investigates the question as to whether an EPS deficient strain of *Bacillus subtilis* can invade or participate in biofilm formation with the wild type, thereby gaining benefits despite

being a non-producer. The authors have established the context for the experiments nicely in the introduction. One question here, regarding the comment that "long term scenarios are difficult to predict", why not undertake such experiments here to increase the novelty and impact? None the less, the results show that repeated subculturing can lead to the evolution of phage producing strains that are ultimately able to displace (kill) non-immune strains (the non-evolved strains) and become a substantial part of the biofilm.

The data in figure S1a are interesting in that it shows that pellicles can be formed where the percentage of the double mutant is as high as 25% while the data in supplementary figures 1d-e, the percentage can reach 50%. Further, the pie charts shown in Figure 2 suggest that in fact, the mutant can increase to 60-75% of the biofilm. Does this mean that there is no 'tragedy of the commons' or does this only apply when the non-producer represents more 75% of the biofilm? This is potentially an important consideration as it perhaps either suggests that there is no issue with the non-EPS producers invading the biofilm or that they are in fact capable of making biofilms, although the data presented and reference literature suggests otherwise.

The second major finding is that the evolved strains produce bacteriophage that are capable of infecting the non-evolved strains (but not themselves). This aspect is likely to be completely separate from the central question posed in the introduction regarding cheaters in the population, but is quite interesting in its own right, and the most novel observation here. There are several statements about the phage and their products that may mediate this effect, but no data presented on how the phage manage to overcome immunity. Presumably, the sequencing data would contain such information and it would seem pertinent that be included here. In this respect, the sequencing data are nicely summarised on a global scale, but there are no details or information on what are those changes and what genes may be affected. For example, were mutations found in the immunity regions of the phage, or in phage receptors which would reduce subsequent infection rates. One very simple explanation is that the phage are evolving into superinfective variants that allow them to efficiently reinfect the unevolved hosts. This effect seems to be independent of whether they are wt or mutant to begin with or the method of selection (cells and spores vs spores only). It would also appear that these wild type strains also acquire similar changes in the phage infectivity. Such dynamics in the host-phage interaction have been shown in other systems, e.g. Lenski and Levin (1985). Constraints on the Coevolution of Bacteria and Virulent Phage: A Model, Some Experiments, and Predictions for Natural Communities. *The American Naturalist*. 125:585-602; Brockhurst et al. (2006). The impact of phages on interspecific competition in experimental populations of bacteria. *BMC Ecology*, 6(1), 19; Brockhurst et al. (2005). The effect of a bacteriophage on diversification of the opportunistic bacterial pathogen, *Pseudomonas aeruginosa*. *Proc. Royal Soc. B-Biol. Sci.*, 272(1570), 1385-1391. and hence some of this literature should be included and used for discussion.

The observations that the variants are more competitive are consistent with publications from Boles et al. (2004). Self-generated diversity produces "insurance effects" in biofilm communities. *Proc. Natl. Acad. Sci.*, 101(47), 16630-16635 and Lee et al. (2016). Interspecific diversity reduces and functionally substitutes for intraspecific variation in biofilm communities. *ISME J*, 10, 846-857, which probably should also be considered here.

Specific comments

If the double mutants do not make pellicles, how are pellicle invasion assays, as shown in figure 3, performed where two sets of double mutants are mixed together?

It would be good to have some more details about the difference in planktonic growth, e.g. did the growth rates differ or just the final amount; what time point is the data in figure 1d represent?

The evolution experiment is very interesting and shows that by transferring only spores, the resulting pellicle can include, in some cases, a substantial proportion of the double mutant. Were inoculation densities similar for both methods (cells + spores vs spores only).

The statement "Spontaneous phage release by the evolved strains and their lytic activity towards the ancestors suggested that higher incorporation of the evolved mutants into pellicles may be the result of newly evolved interference competition through phage weapons." is somewhat confusing as it suggests that the phage are acquiring some proteins or mechanism that facilitates infection or

lysis of the wt. Rather, it seems that what is described is simply evasion of the innate immunity system. this statement should be revised for accuracy and to avoid using unspecific language that becomes confusing. This would apply throughout the the text in using the phrase 'phage weapons'; it sounds cute but is not accurate or clear.

lines 137-140, I think the authors meant that all but two of the lineages showed increased numbers of the double mutant since both lineages 2 and 4 were almost entirely just the wt. while not a critical comment, the description that increase in the mutant was gradual is probably not quite accurate since lineage 4 showed a sudden increase in the mutant on the 8th passage and then by the 10th passage, was again dominated by the wt. line 3 showed the same percentages for the 5th and 8th passage and then suddenly increased by the 10th passage.

Suggest that figure S2 be incorporated into figure 2 since it carries much more information that is important for presenting the data. presumably there is no difference in the percentages for method A, so that portion of figure 2 can remain as presented.

As a small point, it is more direct and probably correct to say that the evolved strains show increased ability to invade wt biofilms or to compete, rather than to say that the wt showed diminished exclusion (which is not what the data show).

lines 241-243 repeat the information from the preceding sentence, so one of those should be deleted.

Figure S6, do the authors intend 'lawn' instead of loan, e.g. a lawn of bacteria tested for plaque formation?

In the title: De novo evolved interference competition promotes the spread of biofilm defectors.

Reviewer #1 (Remarks to the Author):

This is an interesting study in which biofilm-matrix non-producers of *B. subtilis* evolved to more effectively integrate into pellicle biofilms generated by matrix producers. This improved ability to colonize pellicles was associated with reduced population productivity in mixes of one of three evolved non-producers with the ancestral producer. Additionally, increased pellicle colonization, involved a form of phage-related interference competition unique to evolved non-producers in that it was not exhibited by engineered non-matrix-producing mutants of evolved matrix producers. These central results are interesting, novel (to our knowledge) and seemingly sound. However, there are several substantial points that need to be addressed.

Major points

Pellicle disruption

Is it known that the method of biofilm dispersion used prior to dilution plating completely disperses all biofilm cells? Might biofilm cells remain substantially clumped after the dispersion step, but leading to underestimates of population size? Also, might the degree of cell-cell cohesion itself have evolved in the experiment for either producers (MPs for matrix producers), non-producers (NMPs) or both, thus potentially affecting the relative accuracy of dilution plating for the two categories over evolutionary time? Is this potentially a problem for interpreting fitness estimates? If not, please explain why.

We completely agree with the reviewer that low sonication efficiency can lead to underestimation of biofilm population size. Since *B. subtilis* biofilms are the main research focus of our group, we have put substantial efforts to optimize the CFU assay and maximize its accuracy. This was done by testing several sonication conditions (varying the amplitude, sonication time and intervals) and monitoring the sonicated cultures by microscopy to confirm the efficient disruption of biofilm clumps. This allowed establishment of a routine sonication protocol for efficient disruption of 2-3 day-old pellicles of *B. subtilis* 168. The same protocol was used for disruption of the evolved pellicles and its efficiency was confirmed by microscopy. Similar optimization methodology was previously used by other researchers to disrupt pellicles of *B. subtilis* (e.g. Branda et al 2006 Mol Microbiol; Kobayashi and Iwano 2012 Mol Microbiol). Further, the different incorporation possibilities of NMP strains were validated with microscopy of undisturbed pellicles (compare Fig. 1 and Fig. 3).

Pellicle productivity

In lines 212-215, the authors make a general statement that the evolved NMPs (eNMPs) reduce productivity of pellicles. However, this is only in the context of mixes with the WT ancestor, relative to that ancestor alone. Moreover, from Fig. 4 it seems that only one of the eNMPs has a significant negative effect individually, and no test for a general effect across all three eNMPs seems to be reported. But more importantly, the most relevant treatment would seemed to be eNMPs mixed with evolved MPs (eMPS), rather than eNMPs mixed with unevolved MPs because the latter two categories are evolutionarily disconnected from one another. The critical question seems to be whether eNMP +

eMP mixes have lower productivity than NMP + MP mixes. If not, then eNMPs do not lower productivity in the context of the populations in which they co-evolved.

First, we would like to thank the Reviewer for proposing the elegant abbreviations for evolved matrix producers (eMP) and evolved non-producers (eNMP). We believe that these abbreviations will improve the flow of the manuscript. Therefore, we applied these abbreviations in the revised manuscript.

Second, the Reviewer raised an important point to assay the eNMP productivities when combined with eMP strains. We therefore performed additional productivity assays for 3 pairs of eNMP + eMP with common evolutionary histories (B310wtA + B310mA; B410wtB + B410mB; B510wtC + B510mC). The updated Figure 4 is now supplemented with the determined eNMP + eMP productivity data. The data shows that productivity is lower in all 3 pairs compared to the productivity of the ancestor WT, however, the differences are statistically significant only in the cases of B410wtB + B410mB and B510wtC + B510mC mixes. Still, the eNMP + eMP productivities are significantly lower in all combinations, compared to the productivities of the corresponding eMP monocultures indicating that the negative effects of the eNMP strains on entire evolved population is reproducible. These results are now introduced in the results section and figure legend for Fig. 4 was accordingly updated.

Related to this, do the authors have productivity estimates for mixes of eNMPs and eMPs at intermediate time points shown in Fig. S2? If so, it would be nice to see if pellicle productivity correlates negatively with eNMP-frequency in pellicles across all replicates over time.

Again, we appreciate this suggestion based on what we have determined the productivities of the eNMP + eMP populations coming from different evolutionary time points. The pellicles were initiated from frozen fossil records of mixed pellicles from 2nd, 8th and 10th passage. The obtained results are now added as Supplementary Fig. 4B. Although it is difficult to see clear negative correlation between the fraction of the eNMP in the original pellicle and the obtained productivity data, the general trend of decreasing productivity with progressing evolutionary time can be appreciated. The lowest productivity was observed for the pellicle of 10th passage of population 5 that also contained the highest fraction of eNMPs. This is now mentioned in the result section.

Because productivity was measured as biomass production, can the authors say anything about whether how much of the increased productivity of the eMPs is due to increased cell #s vs. to increased matrix production?

We indeed performed both biomass measurements and CFU assay, but, to simplify, only the biomass dataset was presented in the initial version of the manuscript. Now, we provide the corresponding CFU data as a Supplementary Fig. 4A. As evident from the graph, we did not observe increased viable cell numbers in the eMP, as was the case for the overall biomass. We hypothesize that increased frequency of cell lysis, linked to spontaneous lytic induction in the eMP, may lead to release of various substances that contribute to the extracellular matrix and overall biomass. We introduced a reference to Supplementary Fig. 4A in the results and the data is briefly commented in the figure legend of Supplementary Fig. 4.

Mutation rates

The authors do not specify the relevant unit for the mutation rate estimates in Fig. S4C but should do so. Is it per genome, per gene (*rpsL*)? If the latter, the mutation rate would be extremely high, higher in fact than most hyper-mutator strains of *E. coli*. In wildtype *E. coli* strains this rate has been estimated as 5×10^{-10} per target (Levin et al. 2000 GENETICS, 154, 985-997), which would indicate that this *B. subtilis* strain had a mutation rate six orders of magnitude higher than the rate of *E. coli* (which is around 10^{-10} per bp per generation; Lee et al. 2012 PNAS) if the same unit is being considered.

We are grateful for raising this point. The y axis was inaccurately indicated in Supplementary Fig. 4C. The values on the graph correspond to the frequency of streptomycin-resistant mutants (and not to 'rate of mutation' as indicated in the previous version). The figure and its legend have been corrected. It is now also stated in the manuscript that the obtained frequencies are similar as previously reported for other *B. subtilis* strains by Viret & Alonso, 1987.

Minor points.

Nomenclature. The authors frequently use 'community/ies' when the correct term is 'population/s' because all variation in this study is intraspecific.

We appreciate this suggestion and we applied the correct terminology now.

Fig. S1. It would be helpful to show lines corresponding to slope 1.0 in panels A, C, D and E to facilitate rapid interpretation of the data-point distributions.

The slopes are now included as suggested.

Reviewer #2 (Remarks to the Author):

The article suggest that non-pellicle forming *B. subtilis* are excluded from the pellicle, but subsequently evolved to overcome this exclusion by the lytic action of prophage being used as biological weapons. This is a potentially very exciting result, making an important contribution to our understanding of microbial social evolution in biofilms. There is an impressive body of work here, but I had trouble with some of the interpretation of the data.

1. Most significantly, the data as it stands does not demonstrate that the mutant evolves to be more efficient at exploiting the wildtype biofilm. I appreciate that the mutant increases in frequency over 10 transfers, but it may have done this in the absence of genetic change. The key experiments are the competition experiments using the ancestral and evolved mutants and wildtype (Fig 3). These do not show that mutant has become more efficient at exploiting the evolved wildtype (Fig 3A IV), or even that the evolved mutants are better at exploiting the wildtype ancestor (Fig 3A II) but this is probably inevitable given that only 3 evolved mutants and wildtype were tested. Specifically, at least 1 out of 3 mutants doesn't show increased exploitation ability in both cases.

We appreciate this comment, since it encouraged us to re-analyze the pellicle competition dataset and perform additional competition assays to increase the sample size. We repeated the statistical analysis for the competition assays excluding all the samples where initial ratios of the competitors were strongly deviating from 1:1 (values 0.25 and 0.75 were selected as minimal and maximal cut-offs,

respectively). Next, we isolated 2 additional WT (eMP) and 2 additional mutant strains (eNMP) from the evolved populations 3, 4 and 5 (transfer method B) to test them in all combinations of competition assays (evolved mutant vs ancestor WT, evolved WT vs ancestor mutant, evolved WT vs evolved mutant). The analyses revealed that majority of eNMP (6 out of 9) improved in incorporation into pellicle formed by WT ancestor. On the other hand, the ancestor $\Delta eps-\Delta tasA$ performed poorly against all 9 eMPs tested. The improved incorporation also manifested in some of the eMP vs eNMP combinations coming from populations B410m and B510m. We replaced Fig. 3 with an extended version to show all these new data and the figure legend was accordingly modified. The manuscript text was accordingly modified to include the newly performed competition assays to support our claims.

2. I was also very confused with some of the logic. The data convincingly shows that lytic phage are released in evolved population (both wildtype and ancestor). However, the authors argue that this phage release helps exploitation of the biofilm by the mutant in the (co)evolved populations. However, the data in Fig S7 which uses isogenic evolved WT-mutant pairs, both of which release the phage, show no increase in mutant fitness relative ancestral pairs which do not release the phage

Indeed, we do argue that the phage weapons help the evolved mutant to exploit the WT. At the same time, we clearly state that general evolution of phage weapons alone is not improving the exploitation of the eMP by the eNMP. Below we quote some parts of the manuscript where this was clearly stated:

Results section: *'These results indicated that although producers and non-producers showed very similar general adaptation patterns involving major changes in mobile genetic elements, some of these changes were specific to the evolved non-producers, resulting in their improved incorporation into pellicles, most likely through an advantage in interference competition.'*

Discussion: *'(...) the improved incorporation of the non-producers into biofilms was not reproduced when both WT and non-producer strains had identical evolved genetic backgrounds (that of the evolved WT strains). This means that (...) the non-producers are evolutionarily ahead of the producers and carry certain specific changes that allow their improved performance in incorporation into biofilms formed by the evolved WT strains. We believe those specific differences are hidden within prophage elements of the evolved strains, and they could be revealed by de novo sequencing in the future.'*

An elaborate response to this comment follows in point 4 and Review 3/point 2, where we explain the differences in distribution of SNPs between the evolved eMPs and the eNMPs.

3. Sequencing. The number of SNPs seems extremely high to me (<1000 in some cases in less than 100 generations), even given the prophages. Just curious.

We realize that the SNPs numbers are extremely high and unfortunately we are not able to provide a satisfactory explanation for this observation at the moment. At first, we were also skeptical about such high mutation load per relatively short evolutionary time, therefore certain highly-mutated regions were reexamined by Sanger sequencing (to exclude the possibility of errors during Illumina resequencing). In addition the 3 evolved genomes were sequenced 2 times independently (after introduction of the $\Delta eps-\Delta tasA$ mutations the DNA was extracted and sequenced in another batch). The presence of SNPs was confirmed by both methods. As indicated in the materials and methods, the evolved population were sequenced on Life Technologies SOLiD 5500xl sequencer platform, while the isolated clones were sequenced using Illumina MiSeq sequencer and a paired-end approach. The ancestor strains were also

sequenced for reference. Finally, at the same time, using the Illumina MiSeq platform, we have sequenced other *B. subtilis* strains with JH642 and 168 genomic backgrounds that were evolved under different selection regime, but only 5-20 novel SNPs were detected in those samples. Therefore, we are confident on the sequencing platforms.

We may only speculate that high numbers of SNPs could accumulate due to fast/error-prone replication of phage genomes combined with miss-packing and multiple recombination events between different prophage elements. We are hoping that *de novo* sequencing of the evolved genomes which is planned in the future will provide more insights into molecular evolution of these prophage elements.

4. Please be explicit that there are no consistent differences in SNP distribution between evolved mutants and wildtypes

Similar point was also raised by another referee what motivated us to perform detailed analyses of the mutation pattern in the evolved WT and evolved mutant strains. Elaborate response follows in rebuttal to Reviewer 3/point 2.

5. Taken together, the results suggest that the evolution of *B. subtilis* in these conditions results in parallel genomic and phenotypic evolution the mutant and wildtype, with phage release being a notable fitness enhancing trait against ancestral strains. While interesting (and previously observed in other systems, as noted by the authors), this does not seem to have anything specifically to do with the evolution of social exploitation.

We completely agree with the reviewer. We believe that the parallel evolution of prophage elements in WT and mutant was not dependent on social context, but still, it coincidentally changed the social dynamics in the population. Below, we quote some parts of the manuscript where the above idea was emphasized:

Abstract: *'(...) collective behavior can be destabilized as a result of rapid adaptation through mobile genetic elements.'*

Introduction: *'(...) unexpected adaptive events involving mobile genetic elements can shift the social dynamics in the population and destabilize biofilm formation.'*

Discussion: *'(...) newly evolved phage warfare shifts social dynamics in the bacterial population in favor of biofilm non-producers.'* and *'(...) social dynamics in an initially very robust biofilm can be shifted by unexpected evolutionary events.'*

6. On a semantic note, I don't think there is any evidence that the wildtype excludes mutants – it may simply be that the mutants can't exploit the wildtype pellicle. I think this is quite an important distinction both conceptually and mechanistically.

We have to agree that at this point we do not have strong experimental evidence speaking for active exclusion of the biofilm mutants by the WTs. We used the 'exclusion' as a rather symbolic term, but after the reviewers note we realize it may confuse the readers (it might suggest that the WT punishes the mutant) and therefore we decide to remove this expression from the manuscript. The 'exclusion/excluded' was replaced by alternative terms thorough the entire manuscript, for example 'failed to incorporate into the pellicle', 'outcompeted', 'dramatic decline', and the 'diminished exclusion of the mutant' was replaced by 'improved performance of the mutant'.

Reviewer #3 (Remarks to the Author):

The manuscript investigates the question as to whether an EPS deficient strain of *Bacillus subtilis* can invade or participate in biofilm formation with the wild type, thereby gaining benefits despite being a non-producer. The authors have established the context for the experiments nicely in the introduction. One question here, regarding the comment that 'long term scenarios are difficult to predict', why not undertake such experiments here to increase the novelty and impact? None the less, the results show that repeated subculturing can lead to the evolution of phage producing strains that are ultimately able to displace (kill) non-immune strains (the non-evolved strains) and become a substantial part of the biofilm.

1. The data in figure S1a are interesting in that it shows that pellicles can be formed where the percentage of the double mutant is as high as 25% while the data in supplementary figures 1d-e, the percentage can reach 50%. Further, the pie charts shown in Figure 2 suggest that in fact, the mutant can increase to 60-75% of the biofilm. Does this mean that there is no 'tragedy of the commons' or does this only apply when the non-producer represents more 75% of the biofilm? This is potentially an important consideration as it perhaps either suggests that there is no issue with the non-EPS producers invading the biofilm or that they are in fact capable of making biofilms, although the data presented and reference literature suggests otherwise.

This is a very good point that was in fact neglected in the manuscript (as we were attempting to focus on the main story line). As mentioned above, Supplementary Fig. 1 (A, C-E) shows interdependence between the initial and final ratio of the $\Delta eps-\Delta tasA$ (ancestor or evolved) in the pellicle, but it hinders the information about negative effect of the $\Delta eps-\Delta tasA$ load on the total CFUs of the pellicles. We now provide additional charts showing the final ratio of the mutant in the function of total CFU of the pellicle – those are inserted in the upper right corner of the Supplementary Fig. 1A, 1D-F. In all 4 datasets, the negative effects of high mutant load on total CFU can be observed (see negative r values), suggesting that the mutant can impair the population (as typical free-loader) when present in high ratio. However, it is important to remember, that unlike a typical cheater, the $\Delta eps-\Delta tasA$ strain is not able to invade the WT when it starts from low initial frequencies. We added a brief note in the results section to comment on the negative effects of the mutant on fitness: '*Specifically, the mutant showed increased pellicle incorporation, up to $23\% \pm 1.9\%$, which also correlated with decline in total CFUs of the pellicle...*'

Similar question could apply to the evolving mixed populations, which show very high frequencies of the mutant (e.g. 75% in case of population B510, 10th transfer). Following the suggestion of another reviewer, we compared the productivities of pellicles from different evolutionary time points (added as Supplementary Fig. 4B). The biofilm productivities appear to decrease in later evolutionary time points, when the mutant becomes more abundant. This again confirms the negative effect of the high mutant load on the whole population. This experiment was commented in the text:

'In addition, the pellicles formed by the populations from sequential evolutionary timepoints showed overall decreases in productivity in evolutionary time (Supplementary Fig. 4B), presumably caused by the increasing frequency of eNMPs in pellicles (Fig. 2B).'

2. The second major finding is that the evolved strains produce bacteriophage that are capable of infecting the non-evolved strains (but not themselves). This aspect is likely to be completely separate from the central question posed in the introduction regarding cheaters in the population, but is quite interesting in its own right, and the most novel observation here. There are several statements about the phage and their products that may mediate this effect, but no data presented on how the phage manage to overcome immunity. Presumably, the sequencing data would contain such information and it would seem pertinent that be included here. In this respect, the sequencing data are nicely summarised on a global scale, but there are no details or information on what are those changes and what genes may be affected. For example, were mutations found in the immunity regions of the phage, or in phage receptors which would reduce subsequent infection rates. One very simple explanation is that the phage are evolving into superinfective variants that allow them to efficiently reinfect the unevolved hosts. This effect seems to be independent of whether they are wt or mutant to begin with or the method of selection (cells and spores vs spores only). It would also appear that these wild type strains also acquire similar changes in the phage infectivity

Such dynamics in the host-phage interaction have been shown in other systems, e.g. Lenski and Levin (1985). Constraints on the Coevolution of Bacteria and Virulent Phage: A Model, Some Experiments, and Predictions for Natural Communities. *The American Naturalist*. 125:585-602; Brockhurst et al. (2006). The impact of phages on interspecific competition in experimental populations of bacteria. *BMC Ecology*, 6(1), 19; Brockhurst et al. (2005). The effect of a bacteriophage on diversification of the opportunistic bacterial pathogen, *Pseudomonas aeruginosa*. *Proc. Royal Soc. B-Biol. Sci.*, 272(1570), 1385-1391. and hence some of this literature should be included and used for discussion. The observations that the variants are more competitive are consistent with publications from Boles et al. (2004). Self-generated diversity produces 'insurance effects' in biofilm communities. *Proc. Natl. Acad. Sci.*, 101(47), 16630-16635 and Lee et al. (2016). Interspecific diversity reduces and functionally substitutes for intraspecific variation in biofilm communities. *ISME J*, 10, 846-857, which probably should also be considered here.

We find the above suggestion very relevant, therefore we put extra efforts into the comparative analyses of the SNPs in the eMPs and the eNMPs. As the reviewer already suggested, we indeed found some functional differences between the most highly mutated genes in the eMPs and in the eNMPs. First, for each evolved isolate (B310mA, B410mB, B510mC, B310wtA, B410wtB, B510wtC), we listed all mutated CDS together with the corresponding number of SNPs, and function (or hypothetical function) of the gene. The mutated genes were grouped into the following categories: DNA replication/cutting/ligation/repair; Toxins/secretion; Cell wall hydrolysis; Sporulation; Transcriptional regulation; Other; Unknown function (unfortunately, most SNPs were found in genes belonging to this group). This data is now included as a new sheet in Supplementary dataset 1.

We noticed that for the pairs that were sharing an evolutionary history (e.g. B310mA and B310wtA), the total number of SNPs was always higher in the eMP. We now provide a simple linear plots showing all mutated genes (sorted according to their functional group) on the x-axis and the number of SNPs per gene on the y-axis as Supplementary Fig. 5D. Interestingly, the pronounced differences in the number of SNPs between the eMPs and the eNMPs are concentrated to the toxins/secretion-related genes.

Despite the very high number of variables (genes) as compared to the number of samples (3 WTs and 3 mutant strains), we also performed principal component analysis (PCA), hoping to see clustering of the

WTs and the mutants. We included the PCA results as a Supplementary Fig. 5E. As an additional information for the reviewers, a very similar PCA result was obtained when only non-synonymous SNPs were considered in the analysis (probably due to the fact that most of the mutations were synonymous). Although the small number of sequenced genomes does not allow drawing major conclusions from the comparative analysis of SNPs in the eMPs vs. eNMPs, the PCA analysis reveals an interesting pattern that could serve as a good starting point for further studies of the possible mechanism behind a more successful incorporation of the evolved mutants from B410m and B510m populations into biofilms formed by eMPs. Related to the SNPs analysis, the manuscript was supplemented with the following sentences: *'We also compared the distributions of SNPs in the eMPs and eNMPs by analyzing the functions of affected genes. We observed that eMPs accumulated more SNPs than the corresponding eNMPs, especially in genes related to the toxin production and secretion (Supplementary dataset 1, Supplementary Fig. 5DE). However, most of the affected genes belonged to the unknown function category.'*

We appreciate the suggested references that provide additional support to our experimental data. We added the first three references as suggested, however, we believe that we have not sufficient data to support diversity appearance in our experimental system as described in the last two suggested papers, therefore those were not included in the discussion.

Specific comments

If the double mutants do not make pellicles, how are pellicle invasion assays, as shown in figure 3, performed where two sets of double mutants are mixed together?

There must have been a misunderstanding regarding the figure. All competition assays shown in Fig. 3 were performed including the WT (ancestor or evolved) vs $\Delta eps-\Delta tasA$ (ancestor or evolved) co-cultures. We believe that the confusion could originate from the unorthodox way used to present the data, and specifically, by plotting the percent values of each strain in co-culture as box plots in two separate columns (for example 3 A II was showing results obtained for 3 co-cultures: B310mA (1st column) vs WT (2nd column); B410mB (3rd column) vs WT (4th column); B510mC (5th column) vs WT (6th column)). Therefore, we decided to apply a small changes in this figure. Now, the strains in the co-cultures are indicated together (for example as B310mA vs B310wtA) to reduce confusion. Data for the WT is always shown in grey and data for the mutants are in white (as indicated in the legend). We hope the updated version of Fig. 3 is more intuitive than the old one.

It would be good to have some more details about the difference in planktonic growth, e.g. did the growth rates differ or just the final amount; what time point is the data in figure 1d represent?

We expected that the WT is outcompeted by the $\Delta eps-\Delta tasA$ strain in planktonic co-culture due to its lower growth rate. We included the corresponding growth curves of single cultures and calculated their growth rates, see Supplementary Fig. 1C. A brief comment was also included on this in the manuscript text: *'In planktonic culture, $\Delta eps-\Delta tasA$ had a strong fitness advantage over the WT (Fig. 1D) that is likely due to the release of the mutant from the metabolic costs of Eps production¹³, as also indicated by its higher growth rate in planktonic culture conditions (Supplementary Fig. 1C).'*

The evolution experiment is very interesting and shows that by transferring only spores, the resulting pellicle can include, in some cases, a substantial proportion of the double mutant. Were inoculation densities similar for both methods (cells + spores vs spores only).

The reviewer raised an important point, since indeed the selection for spores could accidentally lead to selection bottleneck differences between the two treatments (treatment A – pellicle disruption => transfer; treatment B – pellicle disruption=> heat treatment to select for spores => transfer). In order to maintain similar selection bottlenecks, we previously detected the sporulation frequency in *B. subtilis* pellicles (applying exactly the same conditions as used in the evolution experiment) and adjusted the dilution factors in transfer method A and B, respectively. We supplemented the methods with an explanatory sentence and reference is added to the new Supplementary Fig. 2 that presents the sporulation frequencies: *‘For transfer method A, the mature pellicles were harvested, mildly disrupted, and reinoculated after 100× dilution. For transfer method B, the pellicles were additionally heat-treated after disruption and diluted ×20 during reinoculation. The sporulation frequency in the conditions applied in the evolution experiment was about 20% (Supplementary Fig. 2). To maintain similar selection bottlenecks in the two transfer methods, a fivefold lower dilution factor was used in transfer method B.’*

The statement ‘Spontaneous phage release by the evolved strains and their lytic activity towards the ancestors suggested that higher incorporation of the evolved mutants into pellicles may be the result of newly evolved interference competition through phage weapons.’ is somewhat confusing as it suggests that the phage are acquiring some proteins or mechanism that facilitates infection or lysis of the wt. Rather, it seems that what is described is simply evasion of the innate immunity system. this statement should be revised for accuracy and to avoid using unspecific language that becomes confusing. This would apply throughout the the text in using the phrase ‘phage weapons’; it sounds cute but is not accurate or clear.

The expression ‘phage weapons’ was deleted from the fragment quoted above, and the title of the given results section was adjusted: *‘Phage release from the evolved non-producers facilitates the higher incorporation of mutants into the pellicle.’*

lines 137-140, I think the authors meant that all but two of the lineages showed increased numbers of the double mutant since both lineages 2 and 4 were almost entirely just the wt. while not a critical comment, the description that increase in the mutant was gradual is probably not quite accurate since lineage 4 showed a sudden increase in the mutant on the 8th passage and then by the 10th passage, was again dominated by the wt. line 3 showed the same percentages for the 5th and 8th passage and then suddenly increased by the 10th passage.

The sentence has been modified: *‘Importantly, with several exceptions (in replicates 3 and 4), the percentage of the $\Delta eps\text{-}\Delta tasA$ mutant was increased in each successive passage of these populations (Fig. 2B).’*

Since Figure S2 was combined with Figure 2B (as suggested in a comment below), we slightly altered the text referring to populations from transfer method B, while the data from transfer method A is commented at the end of the paragraph.

Suggest that figure S2 be incorporated into figure 2 since it carries much more information that is important for presenting the data. presumably there is no difference in the percentages for method A, so that portion of figure 2 can remain as presented.

The data has been included in Fig. 2 as suggested by the Reviewer.

As a small point, it is more direct and probably correct to say that the evolved strains show increased ability to invade wt biofilms or to compete, rather than to say that the wt showed diminished exclusion (which is not what the data show).

Similar point was raised by another reviewer, therefore we decided to avoid the 'exclusion' term thorough the manuscript (at this point we are lacking direct evidence that the WT actively excludes the mutant from the pellicle). We replaced 'diminished exclusion of the mutant' by the expression 'improved performance of the mutant'.

lines 241-243 repeat the information from the preceding sentence, so one of those should be deleted.

We were not expressing ourselves clearly enough and indeed the text in lines 241-243 sounded as a repetition of the proceeding sentence. In the first sentence ('more than 50% of SNPs detected in single isolates overlapped with SNPs found in the corresponding populations') we simply wanted to indicate that the isolated strains were a good representative of the corresponding populations. The following sentence is about the similarities between evolutionary unrelated populations/single isolates. In order to make it clear, the second sentence was modified: '*Further analysis of the sequenced genomes revealed that there was also a large parallel mutational overlap between the evolutionarily unrelated populations and single isolates (regardless of WT or $\Delta eps-\Delta tsaA$ background) (Supplementary Dataset 1, Fig. 6AB).*'

Figure S6, do the authors intend 'lawn' instead of loan, e.g. a lawn of bacteria tested for plaque formation?

The mistake was corrected.

In the title: De novo evolved interference competition promotes the spread of biofilm defectors.

Corrected.

Reviewers' comments:

Reviewer #1 (Remarks to the Author):

The authors generally provided effective responses to our comments, but we have few follow-up points that would be good to address.

Pellicle disruption

It's good that the CFU-count optimization experiments were done, but it's not clear that these are described in the Methods. That would be helpful for readers concerned about this point.

Pellicle productivity

The new results examining productivity of eNMPs and eMPs strengthen the manuscript, as do the new Supp. Fig. 4B and the new data distinguishing biomass vs viable cell counts.

Mutation rates

The mutant frequencies are valuable, but tough for readers to readily convert to mutation rates for comparison across studies. Given the many mutations you obtained, it is important to know if you have started your work with a wild type or a hyper-mutator.

Thus, we suggest that you:

A.) document the frequencies in a Table (as was also done by the authors of your cited paper: Viret & Alonso 1987).

B.) Estimate the mutation rates separately from your frequency data with help of maximum likelihood estimation as implemented in the online tool FALCOR (link: <http://www.keshavsingh.org/protocols/FALCOR.html>)

C.) Plot the inferred mutation rates in log-scale.

D.) Discuss the rate in light of reported values of mutation rates.

Line 35: Is 'destabilized' perhaps too strong of a word to describe slight decreases in total-population productivity caused by the evolved non-producers? To many, destabilized would be taken to mean a fundamental undermining cooperation rather than a small decrease in the productivity resulting from a collective behavior.

Reviewer #3 (Remarks to the Author):

NCOMMS-16-23631, Revised

De novo evolved interference competition promotes spread of biofilm defectors

Marivic Martin^{1,*}, Anna Dragoš^{1,*}, Theresa Hölscher^{1,*}, Gergely Maróti², Balázs Bálint³, Martin Westermann⁴, Ákos T. Kovács^{1,#}

Lines 135 to 146. The authors state that there was an increase in the double mutant over time, but this over simplifies the results, which show that this was only true for transfer method B, where spores and not vegetative cells were transferred. The description of the changes in the B cultures also does not quite fit the graph. The authors indicate that replicates 3 and 4 were

exceptions to the observation that there was a successive increase in the mutant over time. However, the data show that 3 actually increased over time, while 4 showed a jackpot at passage 8, but by passage 10, was almost completely wt again. There is no real increase in culture 2 over time either.

Since there is no change for when vegetative cells are transferred, these data suggest that there is something specific to the spores, which may be related to regrowth of the spores or other properties, eg. stickiness etc. This may also be reflected in the sequence data showing a significant concentration of mutations in sporulation related genes for both cell types.

It is indicated that 3 isolates were collected for sequencing from replicates 3-5 (method B), but the data in table S2 has 6 'single isolates' for each replicate. Did the authors intend to say that they collected 3 isolates for each of the wt and mutant morphotypes?

the figure legend for figure 3 does not match the labels in the figure
it seems from the graph that there was no significant difference in the non-evolved double mutant and the evolved mutants from replicates B310B,C as well as B410A or roughly 1/3 of isolates did not increase their proportion of the pellicle (top left panel). The evolved wt strains completely suppress/exclude the ancestral double mutant (top right panel), although this is not described in the manuscript. Then the authors show data for where both evolved populations are used in the competition assays and show that 1/3 of the evolved double mutants do better than the ancestral wt.

The conclusion is that the changes in the evolved double mutants are solely responsible for the effects observed, but I think the comparison in part needs to be between the top left and bottom left panels, where the relative proportions of the evolved double mutants are compared with the wt and evolved wt. For example, B410B (evolved double mutant) is around 25-30% of the population when challenged with the ancestral wt, but 15% when challenged with the evolved wt. This would suggest that the evolved is a better competitor against this evolved mutant than the ancestral wt. similar reductions are observed for other isolates when looking at these two panels.
lines 235

again, replicate B at the 10th transfer has very few evolved double mutants, in contrast to the statement in the text.

For the sequencing data, do the double peaks of the 310wtA strain just mean that the culture is a mixture of the ancestor and the evolved strain?

Figure 6, there are three colours in the bar graph, but only the black colour is explained

While there is a clear trend of increased incorporation when the phage are introduced into naive strains, Figure 7, the increase is very small, e.g. approximately 0.05%, relative to what is shown in figure 3A, where the evolved mutants show around 10% incorporation into the pellicles, which is roughly a 200 fold difference. This would seem to suggest, as indicated by the authors, that the phage may contribute but is not the main driver of this effect. This should probably be removed from the abstract since the relative importance of the phage is actually not clear or is in fact, minor/non-existent. In fact, removing the EPS genes from the evolved mutants appears to have a greater impact than incorporation of the phage. What was the incorporation difference for the ancestral wt and the evolved wt strains without the eps deletion for comparison?

Reviewers' comments:

Reviewer #1 (Remarks to the Author):

The authors generally provided effective responses to our comments, but we have few follow-up points that would be good to address.

Pellicle disruption

It's good that the CFU-count optimization experiments were done, but it's not clear that these are described in the Methods. That would be helpful for readers concerned about this point.

> We introduced a short note about the sonication procedure in the methods section:

“ Prior the CFU assays, pellicles were sonicated according to a protocol optimized in our laboratory (2 cycles each containing 12× 1 sec pulses at 20% amplitude with 1 sec pause between the pulses), that ensured efficient disruption of biofilm clumps (as verified by microscopy) and therefore accurate total cell counts in the pellicles.”

Pellicle productivity

The new results examining productivity of eNMPs and eMPs strengthen the manuscript, as do the new Supp. Fig. 4B and the new data distinguishing biomass vs viable cell counts.

Mutation rates

The mutant frequencies are valuable, but tough for readers to readily convert to mutation rates for comparison across studies. Given the many mutations you obtained, it is important to know if you have started your work with a wild type or a hyper-mutator.

> *Our first thought after obtaining the genome resequencing data was that the presence of a mutator strains causing the observed high mutation frequencies. However, the mutations were clustered suggesting a directed, rather than a general mutation load increase. Further, the ancestor strains, B. subtilis 168 were used in other short and long term evolutionary experiments, where low mutation number was detected during genome re-sequencing. When the same ancestor strain was evolved for ~350 generation in emulsion droplets, 60 SNP and short deletions were identified. In addition, when the ancestor strain was co-evolved together with an eps mutant strain in pellicles for 10 transfer without the sporulation bottleneck, we observed 14 to 95 mutations or short deletions. These unpublished experiments suggest that the used B. subtilis strains has no mutator phenotype.*

Thus, we suggest that you:

A.) document the frequencies in a Table (as was also done by the authors of your cited paper: Viret & Alonso 1987).

> *We have included a table on the mutation frequencies as presented in the cited paper.*

B.) Estimate the mutation rates separately from your frequency data with help of maximum likelihood estimation as implemented in the online tool FALCOR (link:

<http://www.keshavsingh.org/protocols/FALCOR.html>)

C.) Plot the inferred mutation rates in log-scale.

D.) Discuss the rate in light of reported values of mutation rates.

> We tried using FALCOR several times, but failed to obtain mutation rates estimates with the JAVA based web based platform. Therefore, mutation rate estimates was determined using bz-rates web based tool (<http://www.lcqb.upmc.fr/bzrates>), suggested by several our colleagues, who had good and reproducible experience with this platform. The estimated mutation rates were plotted on a log scale. The estimated mutation rates in the evolved strains were comparable to their respective ancestor strain.

Line 35: Is 'destabilized' perhaps too strong of a word to describe slight decreases in total-population productivity caused by the evolved non-producers? To many, destabilized would be taken to mean a fundamental undermining cooperation rather than a small decrease in the productivity resulting from a collective behavior.

> We agree here with the reviewer. We corrected the text as follows both in the abstract ('collective behavior can be destabilized' was changed to 'collective behavior can be disrupted' and in the introduction: (...) shift the social dynamics in the population and destabilize biofilm formation' was changed to 'shift the social dynamics in the population and reduce biofilm formation.'

Reviewer #3 (Remarks to the Author):

NCOMMS-16-23631, Revised

Lines 135 to 146. The authors state that there was an increase in the double mutant over time, but this over simplifies the results, which show that this was only true for transfer method B, where spores and not vegetative cells were transferred.

> To avoid this oversimplification we modified the text as follows:

'Despite the initial incorporation failure of $\Delta eps-\Delta tasA$ from the pellicle (Fig. 1A-C), its representation in the populations was observed to increase',
was changed into

'Despite (...) its representation in certain populations of transfer method B was observed to increase.'

The description of the changes in the B cultures also does not quite fit the graph. The authors indicate that replicates 3 and 4 were exceptions to the observation that there was a successive increase in the mutant over time. However, the data show that 3 actually increased over time, while 4 showed a jackpot at passage 8, but by passage 10, was almost completely wt again. There is no real increase in culture 2 over time either.

> The text was corrected as suggested by the reviewer: 'Importantly, with several exceptions (in replicates 2 that remained relatively stable WT: $\Delta eps-\Delta tasA$ ratio over time and replicate 4 that showed an outlying outburst of $\Delta eps-\Delta tasA$ at passage 8)...'

Since there is no change for when vegetative cells are transferred, these data suggest that there is something specific to the spores, which may be related to regrowth of the spores or other properties, eg. stickiness etc. This may also be reflected in the sequence data showing a significant concentration of mutations in sporulation related genes for both cell types.

> First, we apologize for incorrect annotation of the *yorL* as a spore-related protein. The *yorL* that accumulated over 50 SNPs in the evolved strains encodes for a putative DNA-polymerase (located in

SPbeta prophage region). This gene is now correctly assigned to the DNA-related group in the Supp. Table 1 and in Figure S5D, E.

Indeed, SNPs were detected in a spore-related genes *sspC* (related to protection of spore DNA). Still we believe the *sspC* did not play a major role in the observed phenotypes of the eMPs and eNMPs since the gene was intact in half of the evolved strains and presence of SNPs was independent whether strains were MP or NMPs. We believe that the link between sporulation treatment and evolution of better incorporation of the mutants is linked with phage rearrangements taking part prior sporulation (as natural part of cell development), or possibly with periodic phage induction that could be caused by high-temperature treatment in selection method B - the possible links between sporulation treatment and evolutionary outcomes observed in this manuscript are broadly discussed in the discussion section, however, we have no direct proof on the connection between sporulation and the observed mutations.

It is indicated that 3 isolates were collected for sequencing from replicates 3-5 (method B), but the data in table S2 has 6 'single isolates' for each replicate. Did the authors intend to say that they collected 3 isolates for each of the wt and mutant morphotypes?

> The text was corrected accordingly to make it clear that 3 populations/isolates for each wt and mutant were sequenced.

the figure legend for figure 3 does not match the labels in the figure

> The figure legend was corrected.

it seems from the graph that there was no significant difference in the non-evolved double mutant and the evolved mutants from replicates B310B,C as well as B410A or roughly 1/3 of isolates did not increase their proportion of the pellicle (top left panel). The evolved wt strains completely suppress/exclude the ancestral double mutant (top right panel), although this is not described in the manuscript.

> We believe the Fig. 3A data were fairly commented in the results section, e.g. 'all tested eNMP strains from populations B410m and B510m, and one isolate from population B310m, could increase their fraction within the pellicles as compared to their ancestor when co-cultured with the ancestor WT'. We also commented on the improved performance of the eMP over the ancestral mutant: 'Moreover, the ancestor $\Delta eps-\Delta tasA$ performed even worse when co-cultured with the eMP strains compared to its performance against the WT ancestor (Fig. 3B).'

To emphasize the last result we introduce an additional sentence (inspired by the reviewers comment): 'Therefore the eMPs completely suppressed the ancestral $\Delta eps-\Delta tasA$.'

Then the authors show data for where both evolved populations are used in the competition assays and show that 1/3 of the evolved double mutants do better than the ancestral wt.

The conclusion is that the changes in the evolved double mutants are solely responsible for the effects observed, but I think the comparison in part needs to be between the top left and bottom left panels, where the relative proportions of the evolved double mutants are compared with the wt and evolved wt. For example, B410B (evolved double mutant) is around 25-30% of the population when challenged with the ancestral wt, but 15% when challenged with the evolved wt. This would suggest that the

evolved is a better competitor against this evolved mutant than the ancestral wt. similar reductions are observed for other isolates when looking at these two panels.

> *We completely agree that the eMPs do better in competition against the eNMPs as compared to the WT ancestor. Indeed, this point might have been neglected when discussing the 3A(IV) results. Therefore, we have introduced an additional sentence: 'Overall the eMPs were better at suppressing the eNMPs as compared to the WT ancestor (...)'.*

This observation makes especially sense in the light of further detected, similar changes in the prophage elements of both eMPs and eNMPs. Nevertheless, since eMP + ancestral $\Delta eps\Delta tsaA$ never resulted in improved performance of the mutant, and because certain eMP+eNMP still gave improved incorporation of the mutant (as compared to ancestral combination), we would maintain a conclusion that the evolution of eNMP was responsible for increased incorporation of the mutants into pellicles.

Evolutionary changes specific to eNMP that could potentially explain this effect are depicted on S5D,E.

lines 235 again, replicate B at the 10th transfer has very few evolved double mutants, in contrast to the statement in the text.

> *The text was corrected into: '(...) populations separated from the 10th transfer of method B cultures (replicates 3, 4 and 5), where the frequency of non-producers was observed to increase during evolution (either gradually or periodically), were subjected to (...)'.*

For the sequencing data, do the double peaks of the 310wtA strain just mean that the culture is a mixture of the ancestor and the evolved strain?

> *We exclude this possibility because the DNA for genome sequencing as well as for the Sanger sequencing was isolated from a culture originating from a single colony.*

Figure 6, there are three colours in the bar graph, but only the black colour is explained

> *The legend has been supplemented with explanation (darkness of the bars is proportional to immunity/lytic activity scores values).*

While there is a clear trend of increased incorporation when the phage are introduced into naive strains, Figure 7, the increase is very small, e.g. approximately 0.05%, relative to what is shown in figure 3A, where the evolved mutants show around 10% incorporation into the pellicles, which is roughly a 200 fold difference. This would seem to suggest, as indicated by the authors, that the phage may contribute but is not the main driver of this effect. This should probably be removed from the abstract since the relative importance of the phage is actually not clear or is in fact, minor/non-existent. In fact, removing the EPS genes from the evolved mutants appears to have a greater impact than incorporation of the phage.

> *Our mistake, the y-axis label in Figure 7 was incorrectly indicated, and we apologize for that. The incorporation of the infected $\Delta eps\Delta tsaA$ (ImA, ImB, ImC) is not $\sim 0.05\%$ but $\sim 5\%$, thus the 0.05 value stands for the fraction of the non-producer in the pellicle and not %. To avoid further confusion, we standardized all pellicle-incorporation graphs to match Figure 3 label (results of Figure 7 and Supp. Figure 8 are now shown as relative CFU of non-producer in the pellicle [%]).*

What was the incorporation difference for the ancestral wt and the evolved wt strains without the eps deletion for comparison?

> The incorporation of the evolved WT strains into the pellicle was also increased as can be indirectly deduced from figure S3 showing selection rates for the evolved WT strains. We agree with the reviewer on the importance of this data, therefore we introduce the incorporation values for the evolved WTs in the figure legend of S3 and shortly comment on it in the results section: 'All eMPs showed a positive selection rate and their relative CFU in the pellicle was significantly higher than 50% (Supplementary Fig. 3).'